# The Sec14-like phosphatidylinositol transfer proteins Sec14l3/SEC14L2 act as GTPase proteins to mediate Wnt/Ca$^{2+}$ signaling

Bo Gong, Weimin Shen, Wanghua Xiao, Yaping Meng, Anming Meng*, Shunji Jia*

State Key Laboratory of Membrane Biology, Tsinghua-Peking Center for Life Sciences, School of Life Sciences, Tsinghua University, Beijing, China

**Abstract** The non-canonical Wnt/Ca$^{2+}$ signaling pathway plays important roles in embryonic development, tissue formation and diseases. However, it is unclear how the Wnt ligand-stimulated, G protein-coupled receptor Frizzled activates phospholipases for calcium release. Here, we report that the zebrafish/human phosphatidylinositol transfer protein Sec14l3/SEC14L2 act as GTPase proteins to transduce Wnt signals from Frizzled to phospholipase C (PLC). Depletion of *sec14l3* attenuates Wnt/Ca$^{2+}$ responsive activity and causes convergent and extension (CE) defects in zebrafish embryos. Biochemical analyses in mammalian cells indicate that Sec14l3-GDP forms complex with Frizzled and Dishevelled; Wnt ligand binding of Frizzled induces translocation of Sec14l3 to the plasma membrane; and then Sec14l3-GTP binds to and activates phospholipase Cδ4a (Plcδ4a); subsequently, Plcδ4a initiates phosphatidylinositol-4,5-bisphosphate (PIP$_2$) signaling, ultimately stimulating calcium release. Furthermore, Plcδ4a can act as a GTPase-activating protein to accelerate the hydrolysis of Sec14l3-bound GTP to GDP. Our data provide a new insight into GTPase protein-coupled Wnt/Ca$^{2+}$ signaling transduction.

*For correspondence: mengam@ mail.tsinghua.edu.cn (AM); jiasj@ mail.tsinghua.edu.cn (SJ)

**Competing interests:** The authors declare that no competing interests exist.

## Introduction

Wnt ligands, a large family of secreted lipoglycoproteins, control a large number of developmental events in animals, including cell fate, migration and polarity, embryonic patterning, organogenesis and stem cell renewal (*MacDonald et al., 2009*; *Clevers and Nusse, 2012*). Mammals express 19 Wnt members that bind to a corresponding receptor among 10 Frizzled (Fz) receptors (*Niehrs, 2012*). These receptors have a seven transmembrane span motif characteristic of G protein-coupled receptors, and, following binding of a Wnt ligand, activate different downstream pathways (*Semenov et al., 2007*; *Loh et al., 2016*). In the canonical Wnt pathway, Wnt signaling stabilizes cytoplasmic β-catenin and thereby promotes its nucleus translocation and accumulation to activate downstream target genes transcription (*MacDonald et al., 2009*). Wnts also signal through at least two β-catenin-independent (non-canonical) branches, the Wnt/Planer Cell Polarity (PCP) and Wnt/ Ca$^{2+}$ pathways, during vertebrate development. They are both devoted to modulate cytoskeleton organization to coordinate or polarize cell convergent and extension movements (*Veeman et al., 2003*; *Kühl et al., 2000b*; *Angers and Moon, 2009*). In the Wnt/PCP pathway, the monomeric small GTPases such as RhoA and Rac1 are required for transducing Wnt-Fz signaling to c-Jun N-terminal Kinase (JNK) to direct cell polarity and cell movement (*Huelsken and Birchmeier, 2001*; *Wodarz and Nusse, 1998*).

The Wnt/Ca$^{2+}$ signaling pathway emerged with the observations that overexpression of *Xenopus* Wnt5A and rat Fz2 in zebrafish embryos stimulated intracellular calcium flux and calcium-activated

intracellular pathway (*Kohn and Moon, 2005*; *Slusarski et al., 1997a*, *1997b*). It is demonstrated that pertussis toxin-sensitive heterotrimeric G protein subunits, e.g., Gαo and Gαt, are required for transducing the specific Wnt-Fz-Dishevelled complex signals downstream to activate phospholipase C (PLC) (*Halleskog et al., 2012*; *Halleskog and Schulte, 2013*; *Liu et al., 1999*). The activated PLC cleaves $PIP_2$, a membrane-bound inositol lipid, into diacylglycerol (DAG) and inositol 1,4,5-triphosphate ($InsP_3$). DAG stimulates protein kinase C (PKC) to induce ERK phosphorylation, while $InsP_3$ binds to its corresponding receptor on the ER membrane, opening calcium channels for $Ca^{2+}$ release and increasing cytoplasmic $Ca^{2+}$ ion levels (*Kadamur and Ross, 2013*). Besides regulating cytoskeleton organization, Wnt/$Ca^{2+}$ also can influence CE movements through modulating calcium-dependent cell adhesion or dynamics of calcium storage and release (*Kühl et al., 2001*; *Slusarski and Pelegri, 2007*; *Tada and Heisenberg, 2012*; *Wallingford et al., 2002*). However, until now the interplay between Fz and heterotrimeric G/GTPase proteins in the Wnt/$Ca^{2+}$ signaling pathway is very controversial (*Oner et al., 2013*; *Schulte and Bryja, 2007*). Therefore, it remains mysterious and debated about how to orchestrate the upstream Wnt/Fz stimulation with downstream PLC/$Ca^{2+}$ components.

In recent years, a number of Sec14-like proteins have been identified and characterized. It has been demonstrated that dysfunction of Sec14-like proteins would cause various human diseases, such as breast cancer, prostate cancer, ataxia, and retinal degeneration syndromes (*Cockcroft, 2012*). Sec14-like proteins belong to atypical class III phosphatidylinositol transfer proteins (PITPs) (*Allen-Baume et al., 2002*) and consist of the versatile Sec14 domain associated with a GTPase motif of uncertain biological function (*Habermehl et al., 2005*; *Novoselov et al., 1996*; *Merkulova et al., 1999*). PITPs can transfer phosphatidylinositol (PI) and phosphatidylcholine (PC) between membranes, exchanging PI for PC and vice versa, in order to maintain balanced membrane lipid levels upon consumption of phosphoinositides (*Wiedemann and Cockcroft, 1998*). Although it has been established that PITPs integrate the lipid metabolome with phosphoinositide signaling cascades intracellularly, only very few studies indicate that PITPs can respond to extracellular molecular cues to initiate intracellular signaling (*Xie et al., 2005*; *Kauffmann-Zeh et al., 1995*). So far, there is no evidence about the biological importance of the GTPase motif in the Sec14-like proteins, as well as the crosstalk between PITP family proteins and Wnt/$Ca^{2+}$ signaling.

In this study, we investigated the role of the zebrafish Sec14l3 in embryonic development and the molecular mechanism of its action. We demonstrate that genetic depletion of maternal *sec14l3* results in defects in embryonic convergent and extension (CE) movements by impairing Wnt/$Ca^{2+}$ signaling. Biochemical and genetic data indicate that Sec14l3 can transduce, via its intrinsic GTPase activity, Wnt-Fz signaling to activate phospholipase C.

## Results

### Depletion of maternal *sec14l3* impairs the gastrulation CE movements in zebrafish

We are interested in roles of maternal genes in early development of zebrafish embryos. Our previous RNA-seq data suggested that, among three SEC14-like phosphatidylinositol transfer protein genes (*sec14l1*, *sec14l2*, and *sec14l3*), *sec14l3* is highly expressed in zebrafish eggs. Whole mount in situ hybridization (WISH) confirmed the abundance of *sec14l3* transcripts during cleavage and early blastula stages (*Figure 1A*). Thereafter, *sec14l3* mRNA decreases to undetectable levels by the shield stage and increases again after the bud stage with enrichment in the vasculature cells (*Figure 1A*).

To study *sec14l3* function, we first used two antisense morpholino oligonucleotides (MO), sec14l3-MO1 and sec14l3-MO2, which targeted different sequences around the 5′ untranslated region of *sec14l3* mRNA, to block its translation (*Figure 1—figure supplement 1A*). Since sec14l3-MO1 was more effective than sec14l3-MO2 (*Figure 1—figure supplement 1B*), sec14l3-MO1 was used in subsequent experiments. Compared to standard MO (std-MO)-injected embryos, embryos injected with sec14l3-MO1 displayed slower epiboly in a dose-dependent manner (*Figure 1B*).

To substantiate the knockdown effect, we generated four *sec14l3* mutant lines by targeting the sequence around the translation start codon using transcriptional activator-like effector nucleases (TALENs) technology (*Figure 1—figure supplement 1C*). The first line we obtained was *sec14l3^tsu-*

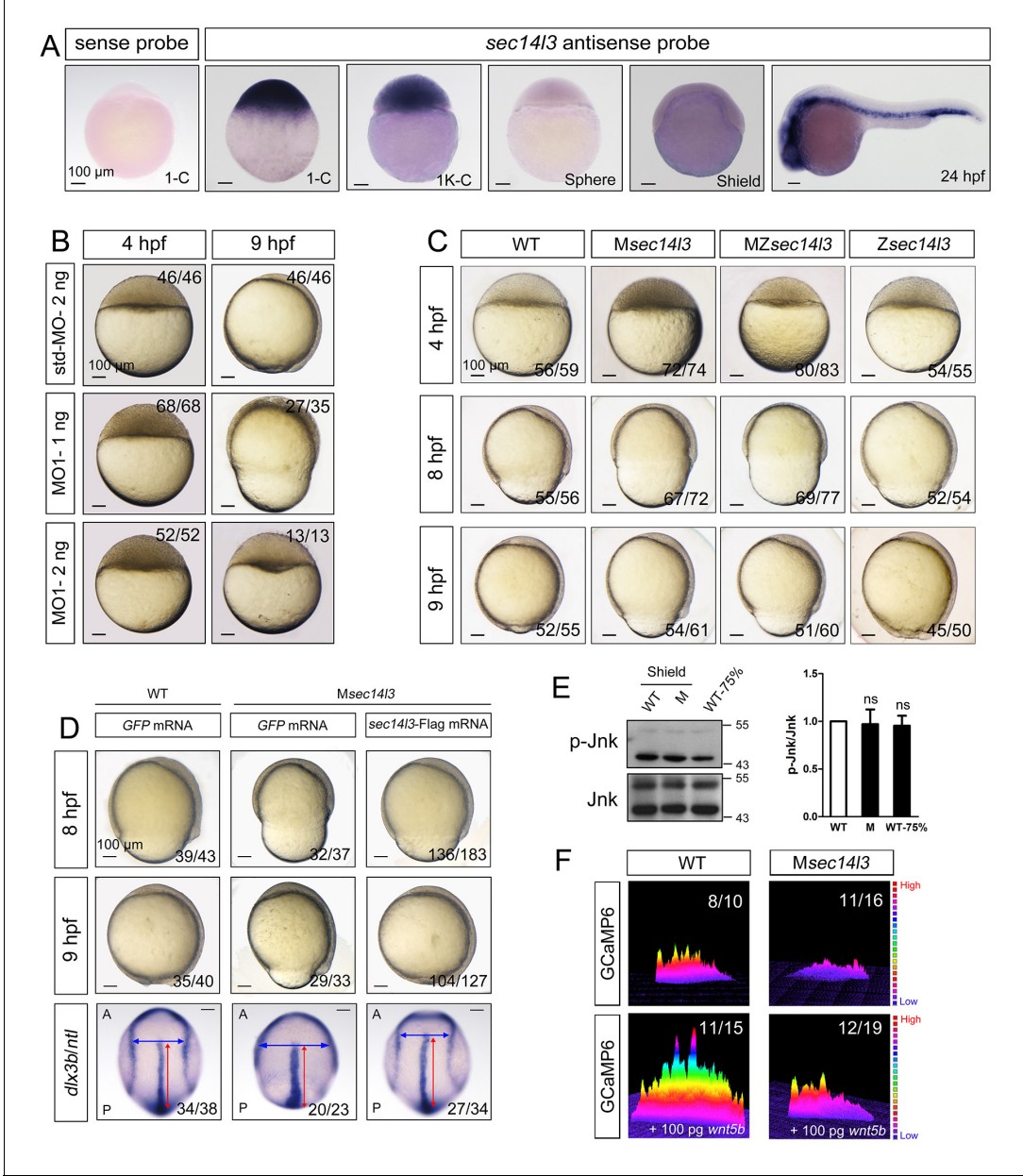

**Figure 1.** Sec14l3 depletion impairs CE movements and Wnt/Ca$^{2+}$ signaling in zebrafish. (**A**) Spatiotemporal expression pattern of *sec14l3*. Embryos were laterally viewed with animal pole to the top or with anterior to the left. Scale bars, 100 μm. (**B**) Morphological defects in *sec14l3* morphants during gastrulation. Scale bars, 100 μm. (**C**) Morphological defects in M*sec14l3*, MZ*sec14l3* and Z*sec14l3* mutants. Scale bars, 100 μm. (**D**) *sec14l3* mRNA rescue assay. 150 pg *sec14l3* mRNA was injected into M*sec14l3* mutants for rescue, and then the morphology and *dlx3b/ntl* marker gene expression were examined. First two panels: lateral views; last panel: dorsal views. Blue and red two-way arrows indicate the width of neural plate and the length of notochord respectively. Scale bars, 100 μm. (**E**) Phosphorylation level of Jnk in M*sec14l3* mutant embryos. p-Jnk (Thr183/Tyr185) and total Jnk were examined at the shield (Morphology comparable) and 75% epiboly stage (Time point comparable) by western blot. Quantification of relative protein levels is shown on the right, represented by mean ± SEM in three separate experiments (see also *Figure 1—source data 1*, ns, non-significant). (**F**) Differential induction of calcium transient activity in zebrafish embryos. Representative calcium release profiles of embryos at the sphere stage in wild-type and M*sec14l3* mutant background with or without *wnt5b* mRNA overexpression. The color bar represents the number of transients: red represents high numbers, blue represents lower numbers, and the peaks represent more active regions. In all panels, the ratio in the right corner indicated the number of embryos with altered phenotypes/the total number of observed embryos.

The following source data and figure supplements are available for figure 1:

**Source data 1.** Numerical data for *Figure 1E*, *Figure 1—figure supplement 1D* and *Figure 1—figure supplement 2B*.

*Figure 1 continued on next page*

*Figure 1 continued*

**Figure supplement 1.** Effectiveness of *sec14l3* MOs and generation of *sec14l3* mutants.

**Figure supplement 2.** Sec14l3 depletion has no effect on cell proliferation or cell cycle progression, as well as cell adhesion between envelop layer (EVL) and deep cells (DC).

**Figure supplement 3.** *SEC14L2* depletion inhibits Wnt/Ca$^{2+}$ transduction in PC3 cells.

$^{td10}$, which carried a 10 bp deletion (−3 to +7). Some zygotic mutants (Z*sec14l3*) of this line survived to adulthood, allowing production of maternal (M*sec14l3*) and maternal-zygotic mutants (MZ*sec14l3*). In two-cell stage M*sec14l3* or MZ*sec14l3* embryos, *sec14l3* transcripts were almost eliminated (*Figure 1—figure supplement 1D*), which was likely due to unstable property of mutant mRNAs. Both M*sec14l3* and MZ*sec14l3* mutant embryos showed a slower epibolic process, which mimicked *sec14l3* morphants, whereas Z*sec14l3* embryos appeared normal (*Figure 1C*). Therefore, the contribution of *sec14l3* to gastrulation cell movements is a strictly maternal-effect. Interestingly, this maternal effect lasted through larval stages as evidenced by a reduced body length in M*sec14l3* mutants compared with control embryos (*Figure 1—figure supplement 1E*). It appeared that cell proliferation and cell cycle progression in M*sec14l3* mutant embryos were unaffected, then we focused on the event of cell movements (*Figure 1—figure supplement 2A*). At the bud stage (about 10 hpf), M*sec14l3* mutant embryos had a broader and shorter embryonic axis, which was marked by the midline marker *ntl* and the neural plate border marker *dlx3b* (*Figure 1D*), indicative of impaired CE movements. Moreover, the defective CE movements of M*sec14l3* embryos were not caused by cell adhesion defects between envelop cell layer (EVL) and deep cells (*Figure 1—figure supplement 2B*) and could be rescued by *sec14l3* overexpression (*Figure 1D*). Maternal mutants of other *sec14l3* lines (*sec14l3$^{tsu-td4}$*, *sec14l3$^{tsu-td8}$* and *sec14l3$^{tsu-td9}$*), which were obtained later on, also exhibited similar phenotypes. Taken together, these data strongly suggest that the maternal, but not the zygotic, contribution of *sec14l3* is critical for normal epiboly and CE movements during gastrulation.

## Sec14l3 is required for Wnt/Ca$^{2+}$ signaling transduction

Since both Wnt/PCP and Wnt/Ca$^{2+}$ can regulate cell movements during embryonic development (*Cha et al., 2008*; *Niehrs, 2012*; *Lin et al., 2010*; *Webb and Miller, 2003*), we then examined which pathway was affected in *sec14l3* mutant embryos. Results showed that Wnt/PCP signaling readout, the phosphorylated Jnk, p-Jnk(Thr183/Tyr185), was almost unaffected in M*sec14l3* mutant embryos, compared to wild-type embryos both at the same developmental stage and time point (*Figure 1E*). Human *SEC14L2*, rather than *SEC14L3*, is expressed in HEK293T and Wnt5-responsive PC3 cells, allowing easier examination of the effect of SEC14-like proteins on related signaling pathways (*Figure 1—figure supplement 3A*) (*Ye et al., 2004*). Like zebrafish *sec14l3*, knockdown of *SEC14L2* in PC3 cells had little effect on p-JNK expression levels (*Figure 1—figure supplement 3B and C*). Therefore, we speculate that Sec14l3/SEC14L2 may not be crucial for the Wnt/PCP signaling pathway.

Next, we used the calcium indicator protein GCaMP6 to visualize calcium transients by confocal microscopy in zebrafish embryos, according to the method reported by Slusarski et al. (*Chen et al., 2013*; *Nakai et al., 2001*; *Slusarski et al., 1997a*). We found that the frequency of calcium release was attenuated obviously in M*sec14l3* mutant embryos either at the basal level or upon Wnt5b stimulation (*Figure 1F*). We therefore conclude that maternal Sec14l3 plays a role in Wnt ligand-dependent calcium release during embryogenesis. Similarly, Wnt5a-induced calcium signal in PC3 cells was also decreased when *SEC14L2* was knocked down by shRNA (*Figure 1—figure supplement 3D*). Thus, Sec14l3/SEC14L2 take part in Wnt/Ca$^{2+}$ signaling transduction by promoting intracellular calcium release.

## Sec14l3 forms complexes with Fz, Dvl and PLC

Fz, Dishevelled (Dvl) proteins and PLC are all implicated in Wnt-induced calcium release (*Kühl et al., 2000a*; *Komiya and Habas, 2008*), but the underlying mechanisms remain elusive. We wondered

whether Sec14l3 could associate with these proteins and performed co-immunoprecipitation (Co-IP) in mammalian cells. We found that Sec14l3 associated with C-terminal of human Fz5 (hFz5-CT) and rat Fz2 (Rfz2-CT) in HEK293T cells (*Figure 2A*, *Figure 2—figure supplement 1A*). The *in vitro*-synthesized hFz5-CT and Rfz2-CT could be pulled down by GST-Sec14l3 (*Figure 2B*). These results support the idea that Sec14l3 directly interacts with Fz proteins. Our domain mapping analysis revealed that the N-terminal CARL-TRIO domain and the C-terminal GOLD domain of Sec14l3 were crucial for its interaction with RFz2-CT (*Figure 2—figure supplement 1A–C*).

Then, we tested physical interaction of Sec14l3 with different Dvl proteins. Co-IP results revealed that Sec14l3 physically interacted with human DVL1, DVL2 or DVL3 in HEK293T cells (*Figure 2C*). Used as a representative, Myc-tagged mDvl2 was determined to directly interact with GST-Sec14l3 (*Figure 2D*). And in MCF7 cells, endogenous SEC14L2 interacted with DVL2 (*Figure 2E*). In addition, this interaction was further validated in zebrafish embryos by overexpressing xDvl2-Myc and Flag-Sec14l3 mRNAs (*Figure 2—figure supplement 1D*). Deletion analysis showed that the Sec14l3's Sec14 domain, but neither the CARL-TRIO nor GOLD domain, was essential for interaction with mDvl2 (*Figure 2—figure supplement 1A,B and E*). On the other hand, the DEP form of xDvl2, consisting of the DEP domain and the C-terminal region of xDvl2, was sufficient for the interaction with Sec14l3 (*Figure 2—figure supplement 1A and F*).

Next, we tested the physical interactions between zebrafish Sec14l3 and Plcδ family members, including Plcδ1a, Plcδ3b and Plcδ4a that are essential for $PIP_2$ hydrolysis into DAG and $InsP_3$. As shown in *Figure 2F*, immunoprecipitation of Flag-tagged Sec14l3 in HEK293T cells retrieved HA-tagged Plcδ4a and Plcδ3b, but not Plcδ1a. Furthermore, Plcδ4a-HA and GST-Sec14l3 fusion proteins were expressed in *E.coli* and purified for pull down assay, which showed a direct interaction between them (*Figure 2G*). Additionally, we also found their interaction in zebrafish embryos (*Figure 2—figure supplement 1G*). Domain mapping analysis revealed that the CARL-TRIO domain and the GOLD domain of Sec14l3, unlike the Sec14 domain, were crucial for interaction with the N2 region of Plcδ4a, including the PH and the EF hand domains (*Figure 2—figure supplement 1A,B,H and I*). Therefore, Sec14l3 utilizes different domains to interact with xDvl2 and Plcδ4a.

To test whether Sec14l3, Fz and Dvl form a complex, we performed two-step Co-IP experiment. Results showed that hFz5-CT-Myc was present in the GST-Sec14l3-Dvl2-Flag complex (*Figure 2H*), suggesting the presence of the hFz5/Dvl2/Sec14l3 ternary complexes. Furthermore, we found that mDvl2 was present in the Plcδ4a complexes, as well as in the Plcδ3b complexes, but absent in the Plcδ1a complexes (*Figure 2I*), which were similar to Sec14l3-Plcδ selective interactions (*Figure 2F*). DVL2 was also proved to interact with endogenous PLCδ4a in MCF7 cells (*Figure 2E*). Moreover, we found that different Wnt ligands stimulation could result in distinct calcium responses. Among of them, Wnt5a had a strong capacity to promote calcium release in PC3 cells (*Figure 2—figure supplement 2A*, and [*Thrasivoulou et al., 2013*]). Upon Wnt5a stimulation, mDvl2-Plcδ4a complex formation could be enhanced in PC3 cells, and knockdown of *SEC14L2* led to a drastic reduction of mDvl2-Plcδ4a complexes (*Figure 2J*), indicating the presence of the mDvl2/Sec14l3/Plcδ4a ternary complexes.

Interestingly, Sec14l3-GFP could be co-immunoprecipitated with Sec14l3-Flag through its CARL-TRIO domain and GOLD domain (*Figure 2—figure supplement 2B and C*), suggesting oligomerization of Sec14l3. It is possible that Sec14-like protein oligomers may facilitate the formation of complexes with Fz, Dvl and Plcδ proteins.

## Sec14l3 is required for PLC-catalyzed hydrolysis of $PIP_2$

Sec14-like proteins are members of PITP and assumed to transfer phosphoinositides (PIs) to the plasma membrane (PM) (*Nile et al., 2010*; *Kearns et al., 1998*; *Wiedemann and Cockcroft, 1998*; *Wirtz, 1991*). To test whether human SEC14L2 and zebrafish Sec14l3 have an effect on PI derivatives accumulation at the PM, we first measured levels of $PIP_2$ in HEK293T cells, the lipid substrate of PLC, using a PH probe, which consists of a GFP-tagged PH domain from PLCδ1 that specifically binds to $PIP_2$ (*Várnai and Balla, 1998*; *Idevall-Hagren and De Camilli, 2015*). If the PI transfer activity of SEC14L2 is blocked, the $PIP_2$ level at the PM should be reduced. However, compared to control cells, transfection of PLCδ1-PH-GFP DNA into HEK293T cells depleted of *SEC14L2* resulted in more PLCδ1-PH-GFP protein in the PM fraction (*Figure 3A*), indicative of more $PIP_2$ at the PM. Confocal imaging also revealed that the PLCδ1-PH-GFP fluorescence at the PM was about 2-fold brighter in *SEC14L2* shRNA stable cells than in the control cells (*Figure 3B*, top panel), which was

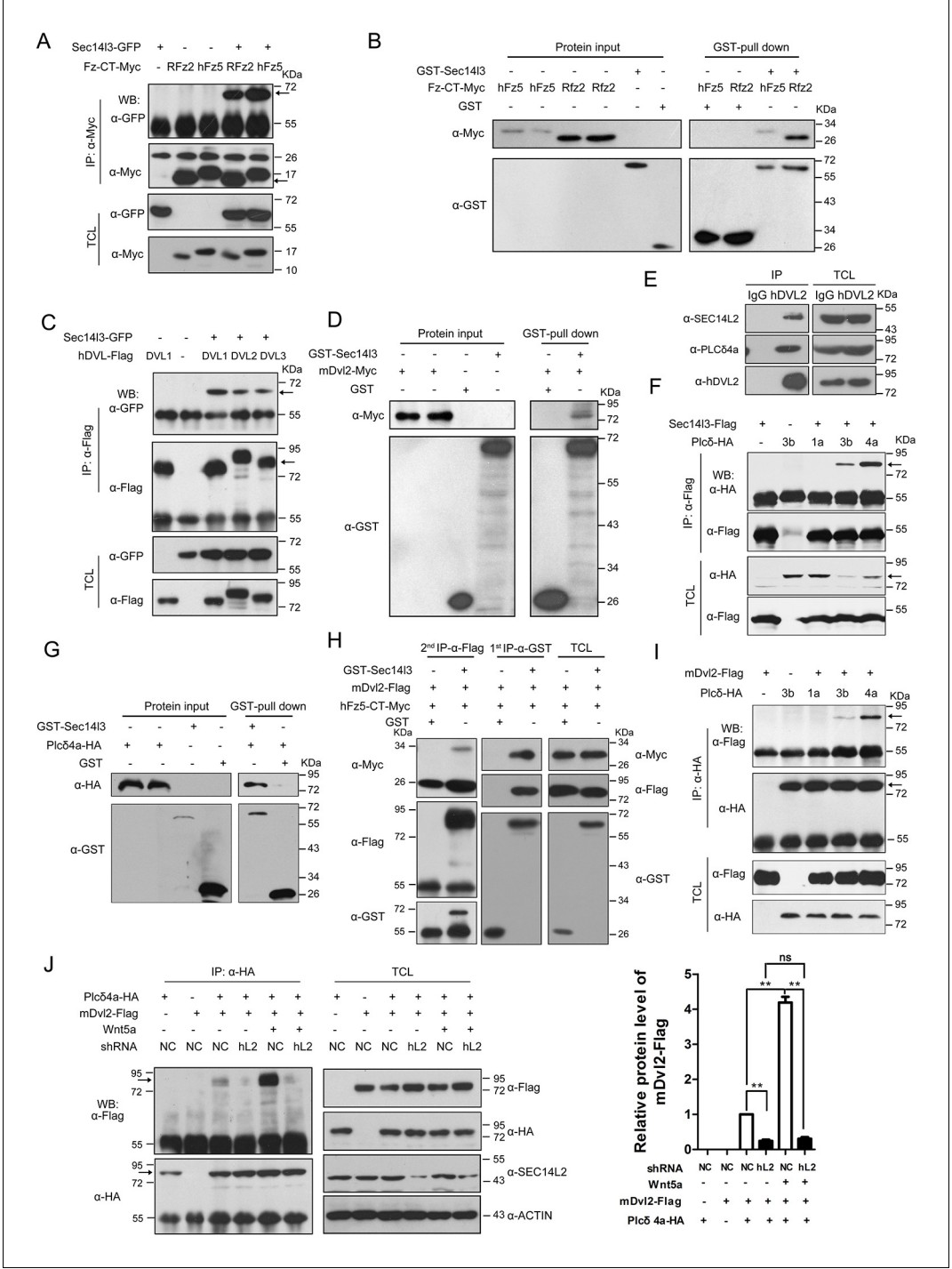

**Figure 2.** Sec14l3 orchestrates complex formation among Fz, Dvl and PLC. (**A**) Sec14l3 interacts with hFz5-CT and Rfz2-CT in HEK293T cells. IP, immunoprecipitation; WB, western blot; TCL, total cell lysates. The target protein in the precipitate was indicated by an arrow (same for other figures below). (**B**) Direct binding of Sec14l3 to hFz5-CT/Rfz2-CT *in vitro*. GST-Sec14l3 and hFz5-CT/Rfz2-CT-Myc were expressed in *E. coli* and purified. (**C**) Sec14l3 was detected in the protein complexes immunoprecipitated with Flag-tagged human DVL1/2/3 from HEK293T cells. (**D**) Direct binding of Sec14l3 to mDvl2 *in vitro*. GST-Sec14l3 and mDvl2-Myc were expressed in *E. coli* and purified. (**E**) Endogenous SEC14L2 and PLCδ4a interacts with DVL2 in MCF7 cells respectively. (**F**) Sec14l3 interacts with Plcδ3b and Plcδ4a in HEK293T cells. (**G**) Direct binding of Sec14l3 to PLCδ4a *in vitro*. GST-Sec14l3 and PLCδ4a-HA were expressed in *E. coli* and purified. (**H**) hFz5-CT, mDvl2 and Sec14l3 form a ternary complex *in vitro*. hFz5-CT-Myc and GST-Sec14l3 proteins purified from *E. coli* were incubated with mDvl2-Flag transfected cell

*Figure 2 continued on next page*

*Figure 2 continued*

lysates. The protein complexes were sequentially pulled down using GST (1 st IP-α-GST) and Flag antibody (2nd IP-α-Flag). Finally the second round immunoprecipitated proteins were detected using the corresponding antibodies. (I) mDvl2 interacts with Plcδ3b and Plcδ4a in HEK293T cells. (J) The interaction between mDvl2 and Plcδ4a is restrained in the stable *SEC14L2* knockdown PC3 cells with or without 400 ng/μl Wnt5a stimulation. Quantification of relative mDvl2 levels from three independent experiments is shown as mean ± SEM on the right (see also *Figure 2—source data 1*, **p<0.01; ns, non-significant).

The following source data and figure supplements are available for figure 2:

**Source data 1.** Numerical data for relative protein level of mDvl2-Flag in *Figure 2I*.

**Figure supplement 1.** Sec14l3 utilizes distinct domains to interact with Dvl2 and Plcδ4a.

**Figure supplement 2.** Sec14l3 can self-assemble into oligomers.

---

consistent with changes of $PIP_2$ levels detected using a $PIP_2$ antibody (*Figure 3B*, lower panel). These results indicate that SEC14L2 may be required for $PIP_2$ hydrolysis rather than for $PIP_2$ transfer.

Then, we switched to detect $PIP_2$ in zebrafish embryos by injecting *PLCδ1-PH-GFP* mRNA. Compared to wild-type embryos, the M*sec14l3* mutant and *sec14l3* morphant embryos accumulated more PLCδ1-PH-GFP/$PIP_2$ at the PM both at 4 and 6 hpf (*Figure 3C and F*, *Figure 3—figure supplement 1A and C*). With regard to $PIP_3$, the metabolic product of $PIP_2$, we also observed a similar PM elevation, indicated by AKT1-PH-mCherry probe, in M*sec14l3* mutants or morphants (*Figure 3—figure supplement 1B and C*). Therefore, consistent with SEC14L2 in mammalian cells, depletion of Sec14l3 leads to the $PIP_2$ accumulation at the PM likely due to inefficient activation of PLC but not deficiency of its PI transfer activity.

PLC-catalyzed hydrolysis of $PIP_2$ produces $InsP_3$ and DAG, two important secondary messengers in cell signaling transduction. $InsP_3$ signals release calcium from intracellular stores, while DAG induces PKC-mediated ERK phosphorylation (*Seitz et al., 2014*). In conjunction with measuring calcium levels (*Figure 1F*), we also measured phosphorylation of ERK (p-ERK) *in vitro* and *in vivo*. As expected, *SEC14L2* knockdown in HEK293T cells caused a reduction of p-ERK (Thr202/Tyr204), but had no effect on phosphorylated AKT (p-AKT Ser473) (*Figure 3—figure supplement 1D*). Similarly, both western blots and immunostaining of zebrafish embryos showed a significant decrease in p-Erk, but not p-Akt in M*sec14l3* mutants (*Figure 3—figure supplement 1E–G*). Collectively, these data establish a role for Sec14l3 in regulation of PLC catalytic activity.

## Wnt/Dvl-induced PLC activation is dependent on Sec14l3

To investigate whether Sec14l3 mediates Wnt5b/Dvl2-induced PLC transduction *in vivo*, $PIP_2$ probe mRNA was injected into blastomeres at the 1 cell stage to visualize $PIP_2$, which was followed by injection of a cocktail of *wnt5b* and *GFP* mRNA into one cell at the 16–32 cell stage to produce mosaic expression of Wnt5b. While injection of *GFP* mRNA alone has no effect on the $PIP_2$ distribution (*Figure 3—figure supplement 2A and E*), $PIP_2$ at the PM in the region with ectopic Wnt5b was significantly reduced compared to in the region without ectopic Wnt5b in the same wild-type embryos (*Figure 3D*, upper panels and *Figure 3F*), suggesting that Wnt5b stimulates PLC-catalyzed $PIP_2$ hydrolysis. However, Wnt5b-dependent PLC activation was obviously inhibited in M*sec14l3* mutant embryos, as evidenced by a much minor reduction in the $PIP_2$ level (*Figure 3D*, lower panels and *Figure 3F*). Like Sec14l3 depletion, *wnt5b* morphant embryos accumulated more PLCδ1-PH-GFP/$PIP_2$ at the PM both at 4 and 6 hpf, compared to std-MO injected embryos (*Figure 3—figure supplement 2B and E*). More importantly, wnt5b-MO-induced PM accumulation of PLCδ1-PH-GFP/$PIP_2$ and CE defects could be individually restored by mosaic and 1 cell stage injection of *sec14l3* mRNA (*Figure 3E,F* and *Figure 3—figure supplement 2C*). Taken together, these epistatic analyses indicate that Sec14l3 can transduce Wnt5 signal to activate PLC in embryos.

Next, we used a truncated form of *Xenopus* Dishevelled, *xDsh-DelN*, to stimulate PLC-catalyzed $PIP_2$. *xDsh-DelN* is a N-terminal DIX domain deletion form that is sufficient to activate the Wnt/PCP and Wnt/Ca$^{2+}$ but not Wnt/β-catenin pathways (*Sheldahl et al., 2003*). As expected, *xDsh-DelN*

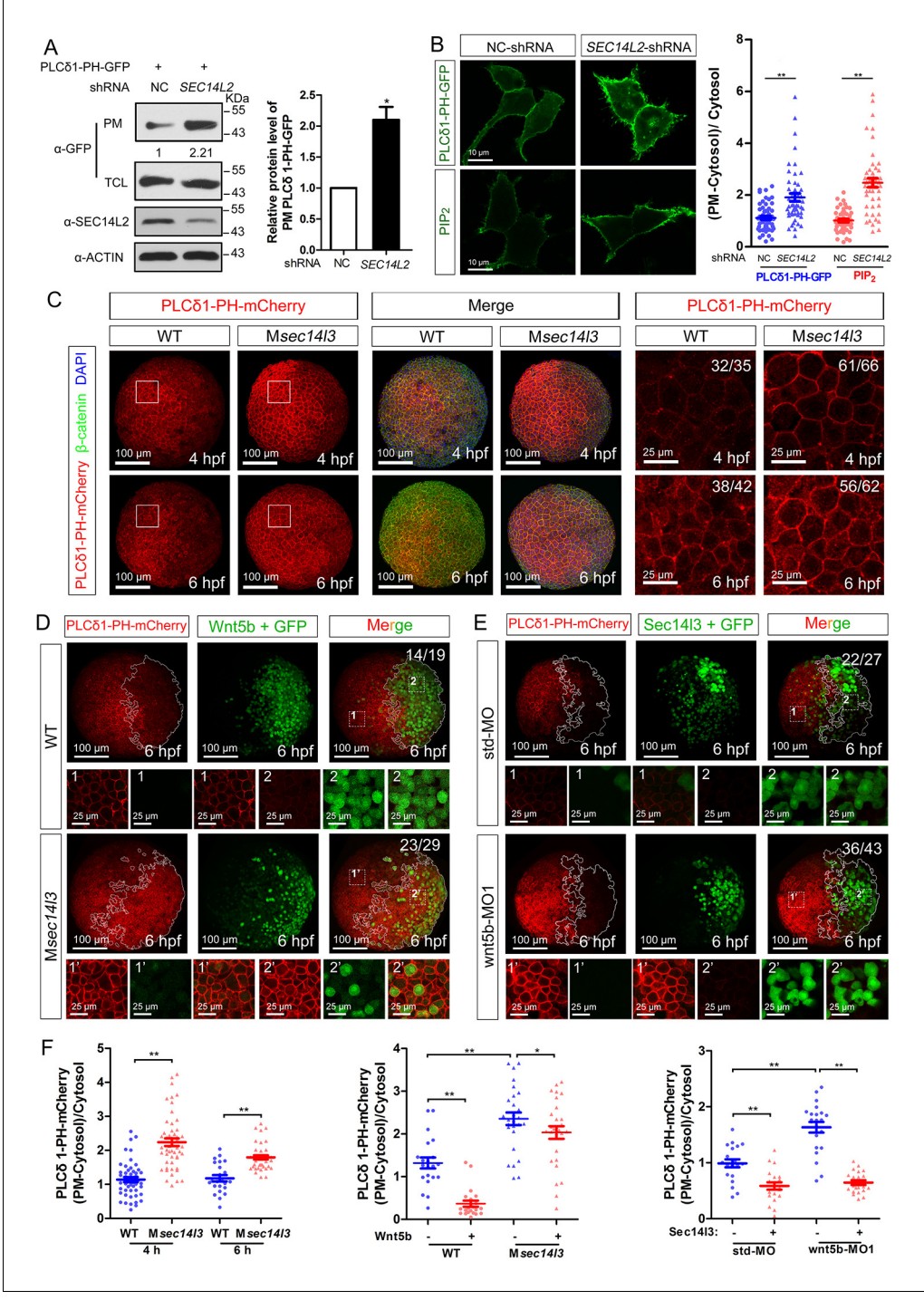

**Figure 3.** Sec14l3 facilitates PLC-catalyzed PIP$_2$ hydrolysis induced by Wnt5b. (**A**) PM isolation analysis of PM PIP$_2$ levels using PLCδ1-PH-GFP as probe in HEK293T cells. Quantification data from three independent experiments are shown as mean ± SEM (see also *Figure 3—source data 1*, *p<0.05). (**B**) Immunofluorescence of PLCδ1-PH-GFP in first panel (transfected with PLCδ1-PH-GFP) and endogenous PIP$_2$ in second panel shows PIP$_2$ accumulation in the PM of stable *SEC14L2*-knockdown HEK293T cells. Data are presented as mean ± SEM (see also *Figure 3—source data 1*, **p<0.01; n ≥ 50 cells from three separate experiments). Scale bar, 10 μm. (**C**) Immunofluorescence of PLCδ1-PH-mCherry (red, PIP$_2$ probe), β-catenin (green, PM marker) and DAPI (blue, nucleus marker) shows PIP$_2$ accumulation in the PM of M*sec14l3* mutant cells. The first two whole embryo panels are 3D views of z-stacks (n = 30 for 4 hpf, n = 34 for 6 hpf), while the last panel is enlarged views of single z-stack pictures (z = 8 for 4 hpf, z = 7 for 6 hpf) from regions encompassed by white boxes. Scale bars, 100 μm for whole

*Figure 3 continued on next page*

*Figure 3 continued*

embryos; 25 μm for the enlarged columns. (**D**) Sec14l3 depletion compromises Wnt5b-induced degradation of PM PIP$_2$. Immunofluorescence of PLCδ1-PH-mCherry (red) and GFP (green, indicating Wnt5b-expressed cells) is shown. Mosaic expression of 100 pg *wnt5b* mRNA was created in embryos with even distribution of *PLCδ1-PH-mCherry* mRNA. White polygons outline GFP expressed cells and single z-stack pictures (z = 10) from numbered regions in the whole embryo panels (3D view of z-stacks) are enlarged. Scale bars, 100 μm for whole embryos; 25 μm for the enlarged panels. (**E**) *sec14l3* overexpression inhibits accumulation of PIP$_2$ in *wnt5b* morphant embryos. Mosaic expression of *sec14l3* by injecting 150 pg mRNA was created in embryos with even distribution of *PLCδ1-PH-mCherry* mRNA in std-MO or wnt5b-MO injected embryos. Single z-stack pictures (z = 11) from numbered regions in the whole embryo panels (3D view of z-stacks) are enlarged. (**F**) PM PIP$_2$ quantification of (**C–E**) by calculating intensity of (PM-Cytosol)/Cytosol PLCδ1-PH-mCherry. Data are shown as mean ± SEM. (see also *Figure 3—source data 1*, \*\*p<0.01; \*p<0.05; ns, non-significant; n ≥ 50 cells from 10 embryos in three independent experiments).

The following source data and figure supplements are available for figure 3:

**Source data 1.** Numerical data for *Figure 3A,B,F* and *Figure 3—figure supplements 1C–E* and *2E*.

**Figure supplement 1.** *sec14l3* depletion results in PIP$_2$ and PIP$_3$ accumulation in the PM.

**Figure supplement 2.** wnt 5b-MO induced phenotypes can be restored by *sec14l3* mRNA.

---

mosaic overexpression also lowered PIP$_2$ levels at the PM in injected clonal region of wild-type embryos (*Figure 3—figure supplement 2D*, upper panels and 2E), while the *xDsh-DelN*- dependent PLC activation was also inhibited in M*sec14l3* mutant embryos (*Figure 3—figure supplement 2D*, lower panels and *Figure 3—figure supplement 2E*). Therefore, it is speculated that Sec14l3 might mediate PLC activation downstream of Wnt5b/Dvl2 stimulation.

## The GOLD and the Gα domains of Sec14l3 are crucial for activating Plcδ activity

To determine which Sec14l3 domain(s) is/are critical for PLC activation, several truncated forms of Sec14l3 were combined with a PIP$_2$ probe for transfection in HEK293T cells. PM isolation assays showed that the truncated forms of CARL-TRIO domain (ΔN) and Sec14 domain (ΔS) still acted similarly to full-length Sec14l3, where a reduction was observed in the amount of PIP$_2$ at the PM (*Figure 4A*). However, rather than leading to PIP$_2$ degradation, forms of Sec14l3 lacking the GOLD domain (ΔG) or Gα subunit (ΔGα) induced PIP$_2$ accumulation at the PM, suggesting that these forms are functioning as dominant negatives (*Figure 4A*). Similar results were observed with immunostaining (*Figure 4B*). To further evaluate this phenotype, we overexpressed the truncated forms of Sec14l3 in human *SEC14L2* knockdown stable cells, and found that enrichment of PIP$_2$ in the PM due to *SEC14L2* depletion was partially compromised by transfecting the full-length form of zebrafish Sec14l3, but not ΔG or ΔGα forms (*Figure 4C*).

In consistent with the above biochemical data, overexpression of *ΔG* or *ΔGα* mRNA in wild-type embryos led to a broader and shorter embryonic axis. And, neither the morphological epiboly defects nor marker gene-labeled CE defects seen in M*sec14l3* mutant embryos were rescued by either of these two truncated mRNAs (*Figure 4D*). Moreover, we found that *ΔG* or *ΔGα* form overexpression caused a significant reduction of p-Erk in wild-type embryos (*Figure 4—figure supplement 1A and B*) and a much lower calcium responsiveness in PC3 cells (*Figure 4E and F*, *Figure 4—figure supplement 1C*), which were quite similar to what happened in *sec14l3* deficient situation (*Figure 1F*). Taken together, these data strongly suggest that the GOLD domain and the Gα subunit domain of Sec14l3 are required for activating Plcδ activity.

## Sec14l3 promotes PM translocation and hydrolytic activity of Plcδ4a in response to Wnt5a stimulation

To further study the mechanism by which Sec14l3-mediated Wnt/PLC activation, we determined whether Sec14l3 translocated to the plasma membrane upon Wnt stimulation. As shown in

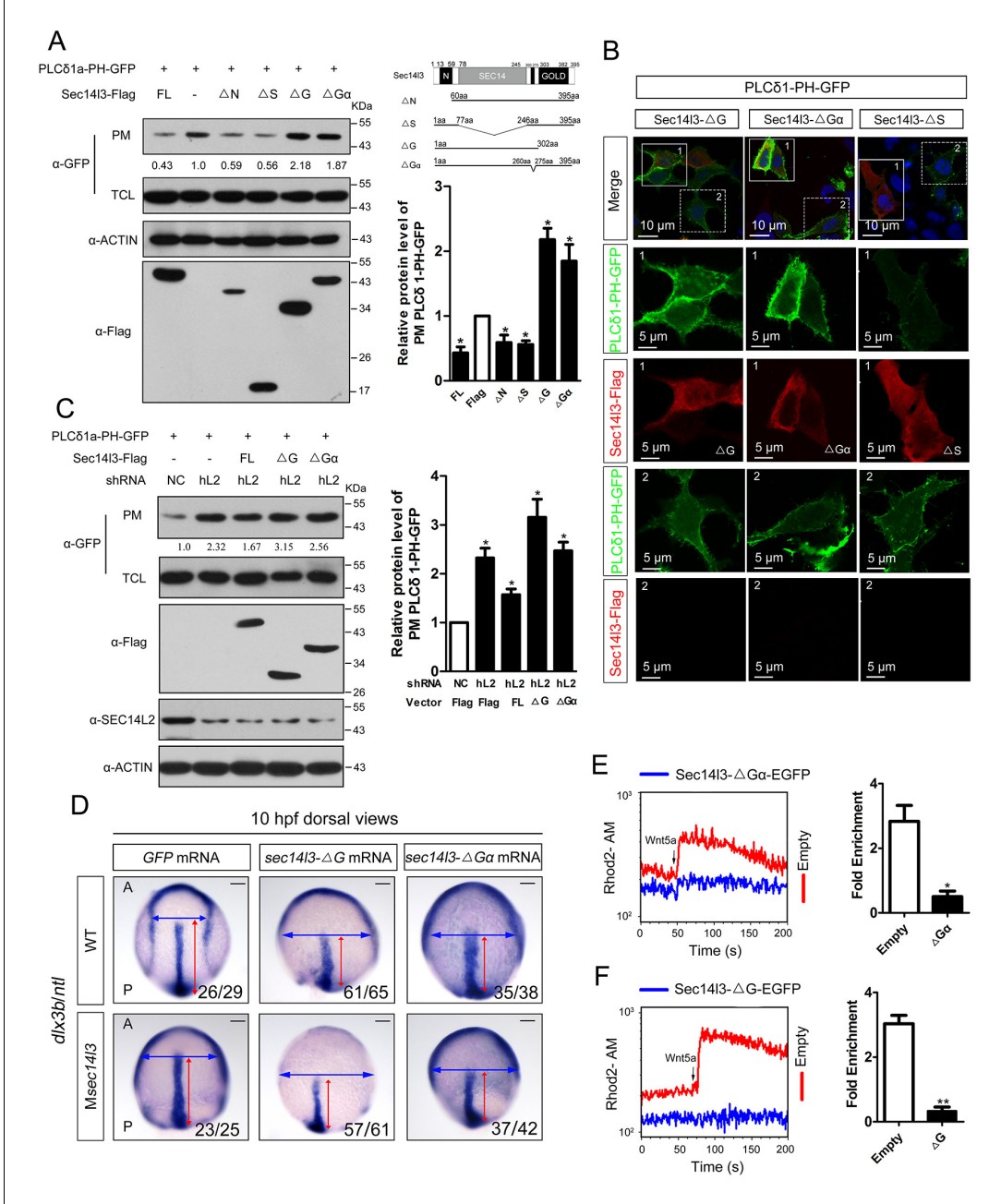

**Figure 4.** Sec14l3 activates PLC dependent on its GOLD and Gα domains. (**A**) Analysis of PIP$_2$ levels in the membrane. Different forms of Sec14l3 (right corner) were co-transfected with PLCδ1-PH-GFP into HEK293T cells, and PIP$_2$-bound PLCδ1-PH-GFP in the PM was detected by Western blot. The relative levels of PLCδ1-PH-GFP in the PM were quantified and presented as mean ± SEM from three independent experiments on the right (*p<0.05). (**B**) Immunofluorescence of PLCδ1-PH-GFP (green, PIP$_2$ probe) in HEK293T transfected with Sec14l3-ΔG, Sec14l3-ΔGα or Sec14l3-ΔS (red) respectively. Regions in white box are enlarged. Scale bar, 10 μm for the first panel and 5 μm for the enlarged panels. (**C**) PIP$_2$ accumulation in stable *SEC14L2*-knockdown cells was not abolished by overexpression of Sec14l3-ΔG or Sec14l3-ΔGα. Statistical data from three independent experiments are presented as mean ± SEM on the right (*p<0.05). (**D**) CE defects in embryos with ΔG and ΔGα Sec14l3 overexpression. *dlx3b/ntl* marker gene expression were examined at 10 hpf after *sec14l3-ΔG* and *sec14l3-ΔGα* mRNA injection respectively. Scale bar, 100 μm. (**E–F**) Flow cytometry of Wnt5a-induced calcium signals in PC3 cells transfected with Sec14l3-ΔGα or Sec14l3-ΔG (blue curves). Left panel shows the kinetic calcium influx over a time course. Right panel shows fold enrichment of calcium influx after Wnt5a stimulation. Data from three independent experiments are presented as mean ± SEM (*p<0.05, **p<0.01). Blue and red curves indicate the transfected and control group respectively. All numerical data represented as a graph in the figure are shown in *Figure 4—source data 1*.

The following source data and figure supplement are available for figure 4:

*Figure 4 continued on next page*

Figure 4 continued

**Source data 1.** Numerical data for relative protein level of PLCδ1-PH-GFP or phosphorylation level ratio in *Figure 4* and *Figure 4—figure supplement 1*.
**Figure supplement 1.** *ΔG* or *ΔGα* form overexpression caused a significant reduction of p-Erk in zebrafish embryos.

*Figure 5A*, *Figure 5—figure supplement 1A*, Wnt5a stimulation induced a rapid translocation of Sec14l3-GFP to the PM, which was similar to the calcium sensor protein STIM1 (*Liou et al., 2005*) and enhanced by co-transfection of Dvl2. To rule out the possibility that Wnt5a-induced Sec14l3 PM recruitment might be a mere consequence of the calcium release, we tested the effect of hFz5/mDvl2 stimulation on Sec14 protein location in HEK293T and MCF7 cells that do not respond to hFz5/mDvl2 for calcium release (*Mikels and Nusse, 2006*; *Foldynová-Trantírková et al., 2010*). In HEK293T cells, endogenous SEC14L2 protein was enriched in the PM by hFz5 transfection, which was further enhanced by hFz5/mDvl2 coexpression (*Figure 5B*). Similar phenomenon was observed in MCF7 cells (*Figure 5—figure supplement 1B*). Additionally, we demonstrated that *SEC14L2* knockdown had no effect on hFz5-mDvl2 interaction (*Figure 5—figure supplement 1C*), which suggests Fz-Dvl interaction might be independent of Sec14-like proteins. Therefore, combining the above data with pairwise biochemical interactions among Sec14l3, Fz and Dvl (*Figure 2*, *Figure 2—figure supplement 1*), we propose that Sec14l3 is a component of the Fz/Dvl/Sec14l3 complex, and its PM recruitment is promoted directly by Fz/Dvl in response to Wnt signaling stimulation.

Then we hoped to know how Sec14l3 regulates PLC activity. PLC has been proposed to serve as a membrane attachment enzyme, which hydrolyzes many substrates such as $PIP_2$ without dissociating from the lipid surface (*Rhee, 2001*). Therefore, we speculated that Sec14l3 might be necessary for recruitment of Plcδ4a to the PM for executing function. To test this hypothesis, we analyzed the subcellular localization of Plcδ4a upon overexpression of Sec14l3. Immunostaining results in MCF7 cells showed that Sec14l3 actually promoted the PM localization of Plcδ4a after co-transfection with hFz5 and mDvl2 (*Figure 5C*). To determine which Sec14l3 domain(s) is/are critical for PLC recruitment, PM isolation assay was performed. In contrast to the cytoplasmic enrichment of Plcδ4a in rest cells, there was a substantial enrichment of Plcδ4a in the PM upon transfection with hFz5 or Rfz2 or both, which was enhanced by co-transfection with *ΔGα* or *ΔG* forms of Sec14l3, but not with *ΔNG* form (*Figure 5D*, *Figure 5—figure supplement 1D*). Furthermore, we found that Rfz2-induced enrichment of Plcδ4a at the PM was inhibited in *SEC14L2* knockdown cells, which could be restored by overexpressing the full-length, *ΔGα* or *ΔG* form, but not *ΔNG* form, of Sec14l3 (*Figure 5E*). Therefore, we speculate that both the TRIO-CARL domain and GOLD domain are required for recruiting Plcδ4a to the PM upon Wnt receptors stimulation. Particularly, different from the receptors stimulation, *ΔG* form is sufficient to block Wnt5a-induced Plcδ4a PM recruitment in PC3 cells, indicating a more important function of the C-terminal GOLD domain of the protein (*Figure 5F*), which is consistent with the functional analysis of $PIP_2$ localization (*Figure 4A*). Taken together, although Wnt5a ligand and its Fz receptors trigger Dvl2/Sec14l3-dependent Plcδ4a PM recruitment in a slightly different way, possibly due to the diverse functions of the C-terminal GOLD2 domain upon ligand stimulation, Sec14l3 is important for the PM translocation of Plcδ4a, which is mainly mediated by the C-terminal GOLD domain rather than the Gα subunit domain.

The next question is whether Sec14-like protein-mediated PM translocation promotes Plcδ4a binding to $PIP_2$. To address this issue, we performed liposome binding assay. As shown in *Figure 5G*, interaction between purified Plcδ4a protein and liposome-bound $PIP_2$ was detected following incubation with HEK293T control cell lysate; however, this interaction was significantly weakened when the *SEC14L2* knockdown cell lysate was used. This result indicates that Plcδ4a accesses $PIP_2$ in a Sec14l3-dependent manner.

## Sec14l3 functions as a GTPase protein in Wnt/PLC activation

We noticed that, although Sec14l3-*ΔGα* acts as a dominant negative form (*Figure 4A–C*), it works differently with *ΔNG* or *ΔG* form in mechanism, based on Plcδ4a PM recruitment results (*Figure 5D–F*, *Figure 5—figure supplement 1D*). As previously reported, human SEC14-like proteins contain a proposed Gα subunit and possess considerable GTPase activity (*Habermehl et al., 2005*; *Novoselov et al., 1996*; *Merkulova et al., 1999*; *Novoselov et al., 1994*). We wondered whether

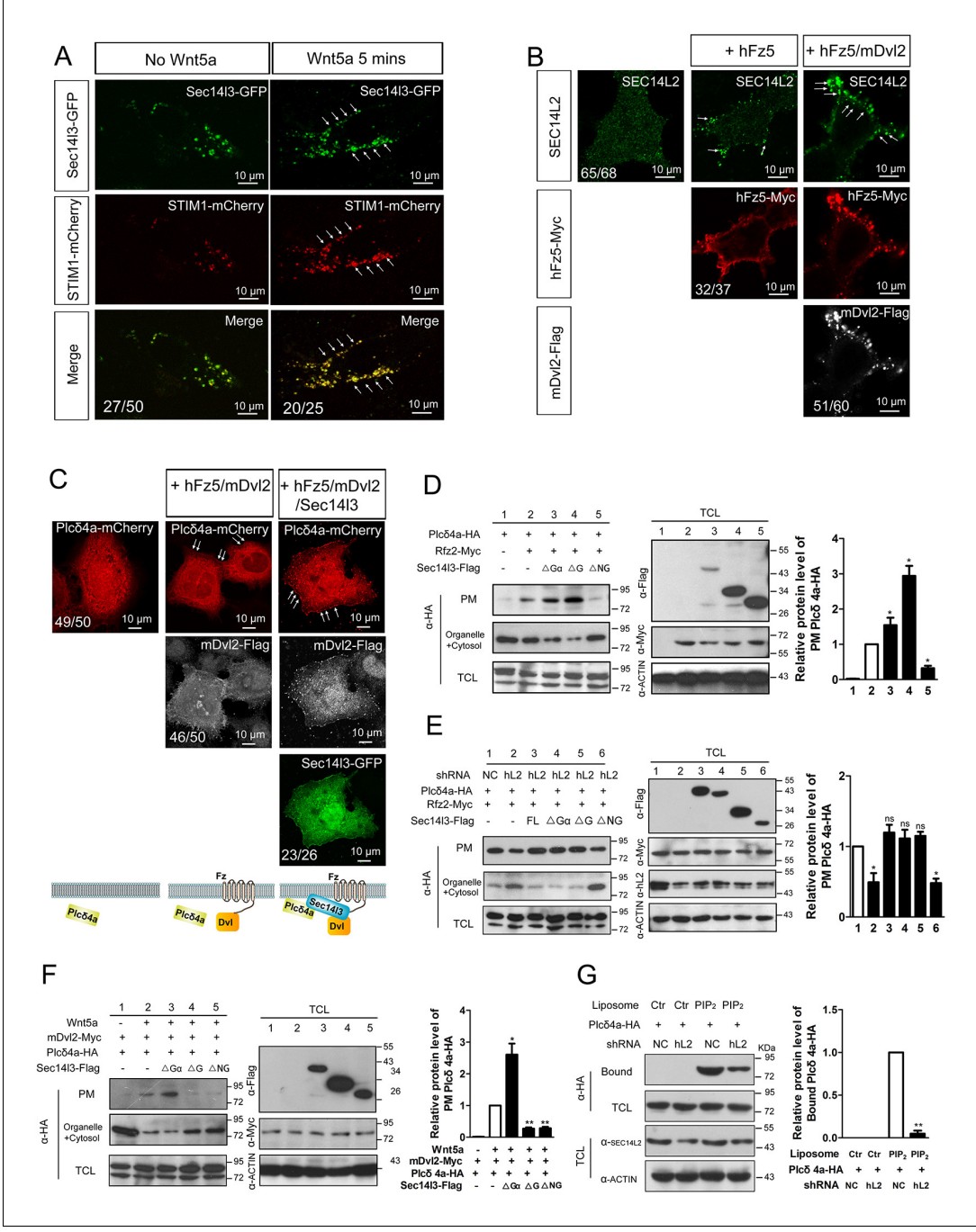

**Figure 5.** PM zone enriched-Sec14l3 recruits PLC for activation upon Wnt5/Fz stimulation. (**A**) Co-localization of Sec14l3 (green) with STIM1 proteins (red) in PC3 cells with or without Wnt5a stimulation. Arrows indicates PM-localized protein after Wnt5a stimulation. Scale bar, 10 μm. (**B**) Immunofluorescence of endogenous SEC14L2 in HEK293T cells with or without hFz5/mDvl2 transfection. Arrows indicates PM-localized SEC14L2 (green). Scale bar, 10 μm. (**C**) Immunofluorescence of Plcδ4a-mCherry (red), mDvl2-Flag (gray) and Sec14l3-GFP (green) in MCF7 cells. PM-localized Plcδ4a (red) are indicated by arrows. The bottom panel show the schematic representation of transfected constructs in corresponding rows and an interpretation of the results. Scale bar, 10 μm. (**D**) The Sec14l3 CARL-TRIO and GOLD domains are important for Rfz2 mediated Plcδ4a recruitment to the PM in the HEK293T cells. Statistical data from three independent experiments are presented as mean ± SEM on the right (*p<0.05; ns, non-significant; same for other statistical data below). (**E**) Rfz2-mediated Plcδ4a PM recruitment is abolished in stable *SEC14L2*-knockdown HEK293T cells and failed to be restored by Sec14l3-ΔNG overexpression. Statistical data are presented. (**F**) The Sec14l3 GOLD domain is important for Wnt5a mediated Plcδ4a recruitment

*Figure 5 continued on next page*

*Figure 5 continued*

to the PM in PC3 cells. Statistical data are presented. (**G**) SEC14L2 depletion perturbs Plcδ4a access to PIP$_2$. Equal amounts of purified Plcδ4a protein and liposomes with or without PIP$_2$ were incubated with control or *SEC14L2* depleted cell lysates. Statistical data are presented. All numerical data represented as a graph in the figure are shown in *Figure 5—source data 1*.

The following source data and figure supplement are available for figure 5:

**Source data 1.** Numerical data for graphs in *Figure 5* and *Figure 5—figure supplement 1*.
**Figure supplement 1.** The subcellular localization of Sec14l3 and its function in the Plcδ4a PM recruitment.

---

zebrafish Sec14l3 has the same property. The high sequence homology in the GTP binding motif and P loop region between zebrafish Sec14l3 and human SEC14L2/SEC14L3/SEC14L4 (*Figure 6A*) suggests a GTPase activity of zebrafish Sec14l3. Then we adopted BODIPY-FL-GTPγS conventional assay and MESG-based single-turnover assay to detect the GTP binding and GTP hydrolysis activities of Sec14l3 (*Lin et al., 2014*; *McEwen et al., 2001*; *Tõntson et al., 2012*; *Webb, 1992*). It estimated the $K_{diss}$ and $K_{hydr}$ rate constants for Sec14l3 to be $0.298 \pm 0.089$ min$^{-1}$ and $0.151 \pm 0.025$ min$^{-1}$ respectively (*Figure 6B–C*). These results indicated that Sec14l3 is a genuine GTPase protein with GTP binding and hydrolysis activities.

A hallmark of G proteins is their ability to undergo conformational switches, from the GDP-bound 'off' state to the GTP-bound 'on' state and vice versa (*Flock et al., 2015*; *Vetter and Wittinghofer, 2001*). To gain full insights into the dynamic switch between Sec14l3 forms, we examined the binding affinity of its interacting proteins with GTPγS/GDP-loaded Sec14l3 *in vitro*. Results indicated that hFz5 had no preference for GDP- or GTPγS-bound Sec14l3, while Dvl2 preferred binding to GDP-bound Sec14l3 (*Figure 6D,E and G*). On the contrary, Plcδ4a showed much stronger interaction with GTPγS-bound form (*Figure 6F–G*). These observations suggest that Dvl2 might participate in the switch of Sec14l3-GDP to Sec14l3-GTP, which then binds to and activates PLC.

Another important question is how active Sec14l3-GTP is cycled back to inactive Sec14l3-GDP. We speculated that Plcδ4a, a binding partner of Sec14l3-GTP, might act as the GTPase-activating protein (GAP). To test this hypothesis, we compared the GTP hydrolysis activity of Sec14l3 in the presence and absence of Plcδ4a protein. Results showed that the $K_{hydr}$ rate was increased about 2.5-fold to $0.315 \pm 0.056$ min$^{-1}$ in the presence of 94 nM Plcδ4a, showing the GAP activity of Plcδ4a (*Figure 6H*). Additionally, quantification of the $K_{hydr}$ constants over different concentrations of Plcδ4a determined its EC$_{50}$ value (50% of maximal effect value) as 21.3 nM (*Figure 6I*). On the other hand, our data disclosed that Plcδ4a is incapable of stimulating GTP uptake by Sec14l3 (*Figure 6J*). Therefore, Plcδ4a acts not only as a Sec14l3-GTP effector but as a terminator, a GAP of Sec14l3-GTP.

To verify the importance of the Gα domain for the GTPase activity, full-length Sec14l3 and Sec14l3-ΔGα were purified from *E. coli*, and resuspended for steady-state GTPase activity assay. Results showed that full-length Sec14l3 stimulated the hydrolysis of GTP in a dose-dependent manner (data not shown), while Sec14l3-ΔGα exhibited the relatively low GTPase activity (*Figure 6K*), indicating the Gα subunit actually engenders the GTPase activity of Sec14l3. Moreover, incubation with lysates from cells transfected with ΔGα form led to much higher levels of Plcδ4a-PIP$_2$ association compared to the full-length form of Sec14l3 (*Figure 6L*); Gα domain deletion had no effect on interaction with hFz5-CT, but impaired its binding to mDvl2 and Plcδ4a (*Figure 6M* and *Figure 6—figure supplement 1A and B*). We speculate that the GTPase activity deficient Sec14l3-ΔGα is unable to bridge Dvl2 and Plcδ4a for complex formation so that Plcδ4a-bound PIP$_2$ may not be hydrolyzed due to PLC autoinhibition (*Hicks et al., 2008*).

## Upregulation of PLC activity rescued *sec14l3* deficiency-induced CE defects in embryos

To confirm that attenuated PLC/Ca$^{2+}$ signaling was responsible for the phenotypes in M*sec14l3* mutant embryos, we used U73122, an inhibitor of PLC (*Ashworth et al., 2007*), to test whether PLC inhibition phenocopies M*sec14l3* mutants. Compared to the DMSO control, U73122 (1.5 µM or 3

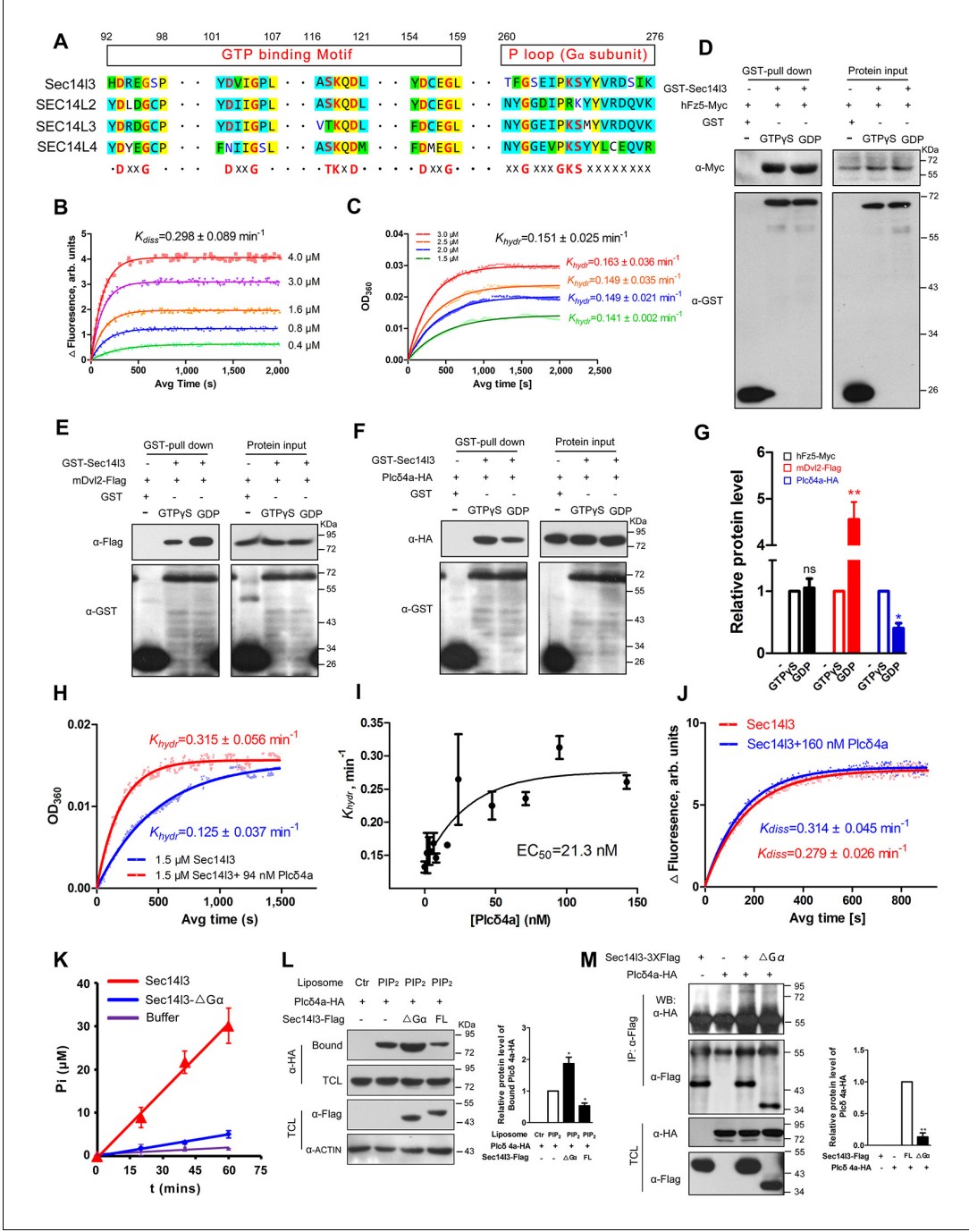

**Figure 6.** Sec14l3 exerts its GTPase activity to prime PLC. (**A**) Protein sequence alignment of zebrafish Sec14l3, and human SEC14L2, SEC14L3, and SEC14L4 in GTP binding motif and P loop region (Gα subunit). Critical amino acids are highlighted in red as consensus at the last panel. (**B**) GTP binding activity of Sec14l3. The fluorescence of BODIPY-FL-GTPγS at indicated concentrations was measured at room temperature ($\lambda_{ex}$ = 490 nm and $\lambda_{em}$ = 510 nm), following the addition of 10 μM Sec14l3. Data are representative uptake curves. The $K_{diss}$ constant of Sec14l3 from three independent experiments is $0.298 \pm 0.089$ min$^{-1}$. (**C**) GTP hydrolysis activity of Sec14l3. Time-course of Pi release from Sec14l3-GTP at indicated concentrations measured by absorbance at 360 nm in the single-turnover assay based on MESG system. Data are representative GTP hydrolysis curves and the $K_{hydr}$ constant of Sec14l3 from three independent experiments is $0.151 \pm 0.025$ min$^{-1}$. (**D**) Full-length hFz5 equally binds to GDP- and GTPγS-bound Sec14l3 *in vitro*. (**E**) Full-length mDvl2 binds preferentially to inactive GDP-bound Sec14l3 *in vitro*. (**F**) Purified Plcδ4a binds preferentially to active GTPγS-bound Sec14l3 *in vitro*. (**G**) Quantification of relative protein level of hFz5-Myc (D), mDvl2-Flag (E) or Plcδ4a-HA (F) bound by GST-Sec14l3 in the GTPγS/GDP form. Data are

*Figure 6 continued on next page*

*Figure 6 continued*

shown as mean ± SEM from three separate experiments (**p<0.01; p<0.05; ns, non-significant). (**H**) Plcδ4a functions as the GAP of Sec14l3. Data are representative GTP hydrolysis curves from three independent experiments. The $K_{hydr}$ constant is statistically significant between two treatments with p<0.05. (**I**) Quantification of the $K_{hydr}$ constants over different concentrations of Plcδ4a. Each concentration is plotted as mean ± SEM from three independent experiments. (**J**) GTP binding activity of 10 μM Sec14l3 in the absence or presence of 160 nM Plcδ4a. Data shows an example of BODIPY-GTPγS uptake curve of three experiments. These two $K_{diss}$ constants have no significant difference by the Student's t test. (**K**) Gα subunit deletion led to a decreased GTPase activity of Sec14l3. Data shown are representative curves out of three replicates and are plotted as mean ± SEM. (**L**) Sec14l3-ΔGα protein overexpression disturbs Plcδ4a-mediated PIP$_2$ degradation *in vitro*. Equal amounts of purified Plcδ4a protein and liposomes containing PIP$_2$ or not were incubated with Sec14l3-transfected cell lysates. Liposome-bound Plcδ4a was eluted and analyzed with quantification data as mean ± SEM on the right (*p<0.05). (**M**) Gα subunit mediates Plcδ4a interaction with Sec14l3 in HEK293T cells. Quantification data from three independent experiments are shown as mean ± SEM on the right (**p<0.01). All numerical data represented as a graph in the figure are shown in *Figure 6—source data 1*.

The following source data and figure supplement are available for figure 6:

**Source data 1.** Numerical data for graphs in *Figure 6* and *Figure 6—figure supplement 1*.

**Figure supplement 1.** Gα subunit deletion had no effect on interaction with hFz5-CT, but impaired its binding to mDvl2.

μM) treatment at the 1 cell stage indeed postponed embryonic epiboly process, phenocoping M*sec14l3* mutants (*Figure 7A*). Additionally, U73122 treatment caused a significant enhancement of PIP$_2$ accumulation at the PM at 4 and 6 hpf (*Figure 7B*), which phenocopied M*sec14l3* mutants.

We explored a possibility to rescue M*sec14l3* mutant phenotype by artificially activating PLC in mutants. We constructed an X/Y linker-truncated form of Plcδ4a, Plcδ4a-Δ28, which was assumed to enhance the basal activity by relieving its autoinhibition (*Hicks et al., 2008*). Injection of *plcδ4a-Δ28* mRNA into M*sec14l3* mutant embryos partially rescued defects in morphology and CE marker gene expression (*Figure 7C*). So it is very likely that Plcδ4a plays a role in gastrulation cell movements by mediating Sec14l3 effect in the Wnt/Ca$^{2+}$ signaling pathway.

## Discussion

To date, only a few biochemical studies based on overexpression or inhibitors of proteins have suggested the implication of heterotrimeric G proteins in Wnt/Ca$^{2+}$ signaling (*Malbon, 2004*; *Katanaev et al., 2005*; *Schulte and Bryja, 2007*). However, it is unresolved as to how Fz/Dvl couples with G/GTPase proteins in Wnt/Ca$^{2+}$ signaling (*Sheldahl et al., 1999*; *Aznar et al., 2015*; *Schulte and Bryja, 2007*). In this study, we show that Sec14-like phosphatidylinositol transfer proteins can function as GTPase proteins in Wnt/Ca$^{2+}$ signaling. As modeled for Sec14l3 (*Figure 7D*), Sec14l3-GDP can form complexes with Fz and Dvl; in respond to non-canonical Wnt stimulation, activated Sec14l3-GTP associates with and activates PLC at the PM, and promotes PLC-mediated PIP$_2$ hydrolysis to generate second messengers that propagate the Wnt/Ca$^{2+}$ signaling cascade. In zebrafish, depletion of maternal *sec14l3* impairs Wnt/Ca$^{2+}$ signaling transduction and consequently causes defective gastrulation cell movements. Our findings not only reveal a critical function for Sec14l3 in regulating Wnt/Ca$^{2+}$ signaling, but provide a comprehensive view of mechanisms about GTPase proteins involvement during the signaling transduction, breaking the argument whether the 7-TM Fz can directly bind to and activate G proteins. We propose it is the GTPase proteins, Sec14l3/SEC14L2, other than the classical heterotrimeric G proteins, that can simultaneously bind to upstream Fz/Dvl and activate downstream PLC for signal propagation. Therefore, the function of Sec14-like proteins is not limited to regulate the exchange of membrane lipids.

The direct Fz-Sec14l3 interaction can also be interpreted as Sec14l3-mediated regulation at the receptor level, such as Fz internalization or recycling in Wnt/Ca$^{2+}$ signaling (data not shown), which needs to be further investigated. Although our clues so far can't discriminate the transducer as a trimer or multimer, the organizer function of Sec14l3 in Wnt/Ca$^{2+}$ is recognizable. What's more, it has

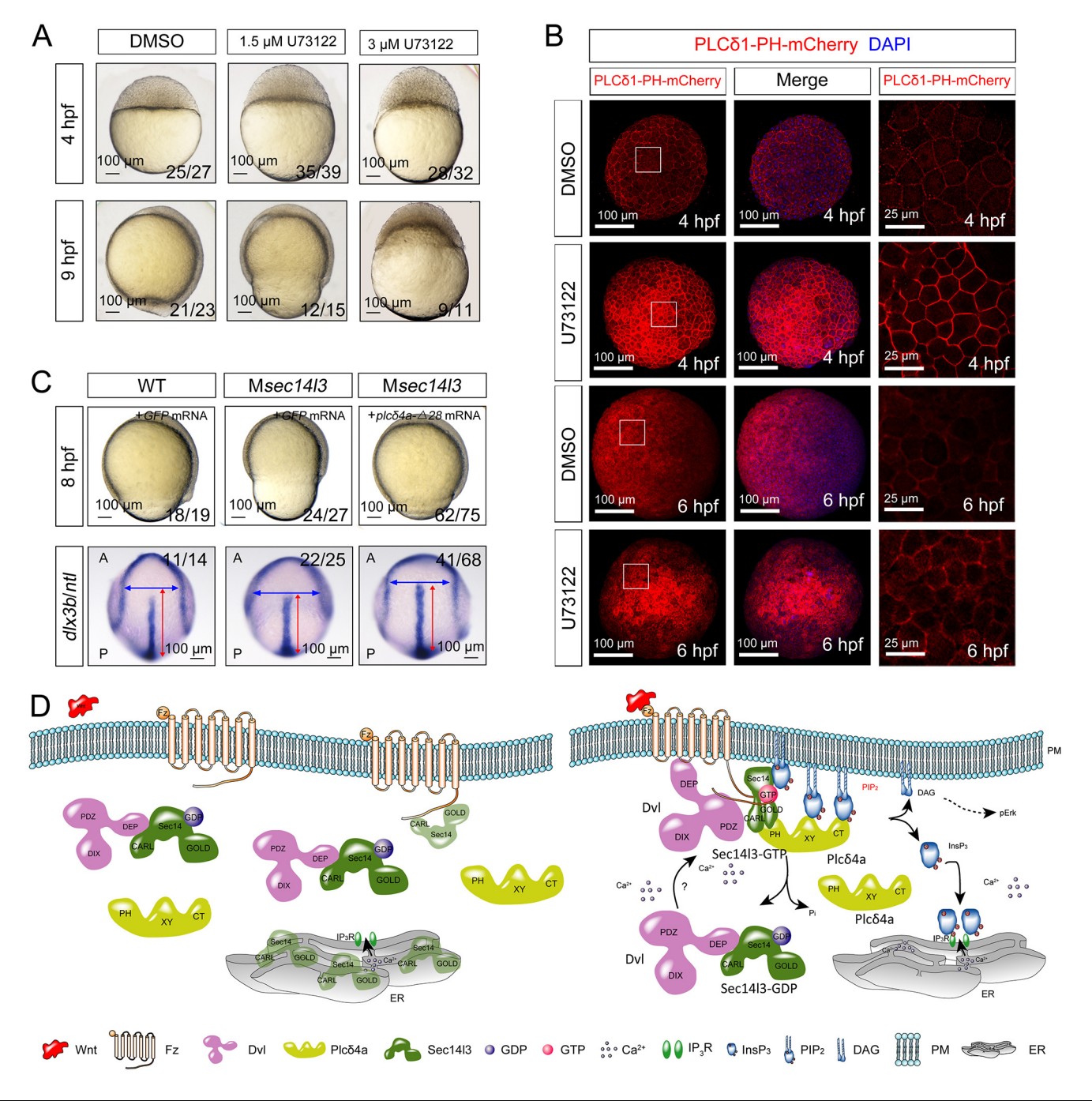

**Figure 7.** Sustained PLC activity partially rescues M*sec14l3* defects. (**A**) Morphological defects in zebrafish embryos treated with 1.5 µM or 3 µM U73122 from the 1 cell stage. DMSO treated group serves as a control. Scale bars, 100 µm. (**B**) Confocal imaging of PLCδ1-PH-mCherry (red, PIP₂ probe) shows the PM accumulation of PIP₂ in zebrafish embryos treated with 1.5 µM DMSO or 1.5 µM U73122 at 4 and 6 hpf. The first two whole embryo panels are 3D views of z-stacks, while the last panel is enlarged views of single z-stack pictures (z = 5 for both 4 hpf and 6 hpf) from regions encompassed by white boxes. Scale bars, 100 µm for the whole embryos; 25 µm for the enlarged columns. (**C**) Active Plcδ4a overcomes the CE defects in M*sec14l3* mutants. 100 pg *plcδ4a-Δ28* mRNA was used for injection. Lateral views for embryos in the first panel, dorsal views for those in the last panel. Blue and red two-way arrows indicate the width of neural plate and the length of notochord respectively. Scale bars, 100 µm. (**D**) Hypothetic working model of Sec14l3 participation in Wnt/Ca²⁺ signaling. In the absence of Wnt (left panel), Sec14l3 is mainly maintained in the ER and cytoplasm, forming a heterodimer with Dvl in its inactive state, Sec14l3-GDP. Upon Wnt5 stimulation (right panel), Fz/Dvl-mediated Sec14l3 is recruited to the PM and switched to the active state, Sec14l3-GTP, and subsequently promotes Plcδ4a localization from cytoplasm to the PM and then the consequent activity at least in two

*Figure 7 continued on next page*

*Figure 7 continued*

aspects: its PIP$_2$ hydrolytic activity to generate second messenger InsP$_3$ and DAG for signaling propagation (Ca$^{2+}$ release and p-Erk activation); and its GAP activity to terminate Sec14l3-GTP.

been suggested that lipid transfer proteins may not simply function as diffusible vehicles mediating lipid transfer between membranes, but also devices of assigning PI lipids to various enzymatic reactions in a strictly regulated biological context (*Mousley et al., 2012*). Our findings disclose multifaceted functions of lipid transfer proteins such as Sec14l3 and fill an important gap in our understanding of how Wnt/Fz/Dvl transduce the signal to PLC for Ca$^{2+}$ release.

Although Sec14l3 is initially identified as a member of PITP, its depletion does not cause PIP$_2$ or PIP$_3$ reduction at the PM both in zebrafish embryos and mammalian cells. Our studies demonstrate that the depletion of Sec14l3 leads to PIP$_2$ accumulation at the PM due to inefficient activation of PLC. However, we cannot exclude the possibility of the involvement of Sec14l3 PI transfer activity in embryonic development because other PITP family members, such as Sestd1 and Sec14l1, are also highly expressed in zebrafish embryos (data not shown). Therefore, to investigate the intrinsic transfer activity of Sec14l3 during early embryogenesis, genetic analysis of double or triple mutants is necessary.

One of the most important aspects of our work is the discovery that the GTPase activity of Sec14l3 is critical for Plcδ4a activation. We find that Plcδ4a binds to Sec14l3-GTP with apparently higher affinity and functions as a Sec14l3-GAP protein. However, Sec14l3-GEF proteins remain unknown. Wnt receptor Fz is a kind of G protein-coupled receptors, which can act as GEF proteins for their cognate G proteins upon binding of a ligand (*Malbon, 2004*; *Schulte and Bryja, 2007*; *Dijksterhuis et al., 2014*). We doubt that hFz5 or Rfz2 functions as the Sec14l3-GEF protein, because these Fz proteins show similar binding affinity towards GDP- or GTP-bound Sec14l3 and cannot accelerate the GTP uptake by Sec14l3 (*Figure 6D,G* and *Figure 6—figure supplement 1C*). Considering that Fz receptors can activate Gαi proteins and enhance Wnt/PCP signaling via the Dvl-binding protein, Daple, a novel non-receptor GEF (*Aznar et al., 2015*; *Ishida-Takagishi et al., 2012*), and Dvl2 does prefer to bind toward Sec14l3-GDP rather than Sec14l3-GTP in our hand (*Figure 6E and G*), we tend to believe that Dvl2 serves as a scaffolding protein to recruit an unknown Sec14l3-GEF, thereby enhancing Sec14l3-GTP formation. As for which is the particular GEF for Sec14l3 in Wnt/Ca$^{2+}$ signaling transduction, more studies are needed. Besides, with the aid of available configuration-specific Sec14l3-GTP antibody in the future, it will be of great interest to figure out the conformational switches *in vivo*.

In summary, this study sheds light on a unique feature of the Sec14l3 protein. Through its intrinsic GTPase activity, it is capable of tightly coupling phospholipase activation with the proximity of PLC to its substrate. Moreover, these findings also provide the mechanism by which Dvl promotes calcium signaling.

## Materials and methods

### Embryos, injection and TALEN mutants generation

*sec14l3* TALEN mutants were generated in the *Tg(flk:EGFP;gata1:dsRed)* (PRID:ZFIN_ZDB_-FISH_150901_14755; ZFIN_ZDB_ALT_051223_6) transgenic fish using the FastTALE TALEN Assembly Kit (SiDanSai, Shanghai, China). The target site was near the start codon and included 22 bp both upstream and downstream (*Figure 1—figure supplement 1C*). To identify the candidate fishes with mutated alleles, genomic DNA was extracted from the tail and amplified using primer pairs as follows: the forward primer 5'-ccagcggcggagataaatc-3' and the reverse primer 5'-acatctatgacaga-cagcaatg-3'. The amplicons were purified for sequencing to determine the mutation types or digested with NlaIII to distinguish wild type and mutant embryos. The progenies derived from crosses between *secl14l3* heterozygotes were raised to adulthood, and homozygous mutant males and females were identified by genotyping. MZ*sec14l3* or M*sec14l3* mutant embryos were obtained by crossing homozygous mutant females to homozygous mutant males or wild-type males, respectively; Z*sec14l3* mutant embryos were obtained by crossing heterozygous females to heterozygous

males. Fishes were handled according to the institutional animal care and use committee (IACUC) protocol (AP#13-MAM1), which was approved and permitted by the Tsinghua University Animal Care and Use Committee. Embryos were staged according to *Kimmel et al. (1995)*.

## Constructs

Zebrafish *sec14l3* and human *SEC14L2* full coding sequences were amplified using the following primers: *sec14l3* with 5'-CCGGAATTCATGAGCGGAAGGGTTGGAGATC-3' (forward) and 5'-CCGC TCGAGCTAGTTGTCTGATTGGTTGAC-3' (reverse), and *SEC14L2* with 5'-CCGGAATTCA TGAGCGGCAGAGTCGGCGATC-3' (forward) and 5'-CCGCTCGAGTTATTTCGGGGGTGCCTGCC-3' (reverse). For information on the other constructs used in this study, please refer to the *Supplementary file 1*.

mRNAs, morpholinos, and microinjection mRNAs were synthesized from corresponding linearized plasmids *in vitro* using a mMessage mMachine kit (Ambion/Thermo Fisher Scientific, Waltham, MA) and purified with RNeasy Mini Kit (QIAGEN, Duesseldorf, German). Morpholinos were synthesized by Gene Tools, LLC. The sequences of MOs used in our study are as follows: sec14l3-MO1, 5'-TCAGATCTCCAACCCTTCCGCTCAT-3'; sec14l3-MO2, 5'-ATGTCGCCACGAGTGCAGCAGAAAT-3'; wnt5b-MO1, 5'-GTCCTTGGTTCATTCTCACATCCAT-3'; and std-MO, 5'-CCTCTTACCTCAGTTA-CAATTTATA-3'. About 1–1.5 nl of mRNA (or morpholino solution) was injected into the yolk at the 1 cell stage for ubiquitous expression or into one single cell at the 16–32 cell stage for clonal expression using the typical MPPI-2 quantitative injection equipment (Applied Scientific Instrumentation Co., Eugene, OR). The injection dose was the amount of the mRNA or morpholino received by a single embryo.

## Whole-mount in situ hybridization and immunofluorescence

Whole-mount in situ hybridization was performed essentially following a standard protocol. Digoxigenin-UTP-labeled antisense RNA probes for detecting *dlx3b* and *ntl* mRNA were generated *in vitro* using a linearized plasmid (*Huang et al., 2007*). Following in situ hybridization, the embryos were immersed in glycerol and photographed using the Ds-Ri1 CCD camera under a Nikon SMZ1500 stereoscope. Embryonic immunofluorescence was carried out as previously described (*Zhang et al., 2012*), and the following antibodies were used: rabbit anti-p-Erk (Thr202/Tyr204) (Cell Signaling Technology, Danvers, MA, #9101, RRID:AB_331646), rabbit anti-p-Akt (Ser473) (Cell Signaling Technology, #4060, RRID:AB_2315049), mouse anti-SEC14L2 (ORIGENE, Rockville, MD, TA503723, RRID: AB_11126641), phalloidin (Sigma, St. Louis, MO, p1951, RRID:AB_2315148). Immunostained embryos were imaged under Nikon A1RMPSi lasers scanning confocal microscope using z-stack devices with 3.5 µm interval. And whole embryo images are 3D views of z-stacks, while the magnified images are pictures of a single z plane snap.

## Cell culture, transfection and stable cell line establishment

HEK293T cells (RRID:CVCL_0063) were cultured in DMEM (Life Technologies) supplemented with 10% FBS (Hyclone, Logan, UT) and 50 µg/ml penicillin/streptomycin (PS) (Invitrogen). PC3 cells (RRID:CVCL_0035) were cultured in F12K with 10% FBS and 50 µg/ml PS. All cell lines were obtained from the Cell Resource Center, Peking Union Medical College (which is the headquarters of National Infrastructure of Cell Line Resource, NSTI). Cell lines were checked for free of mycoplasma contamination by PCR and culture. Its species origin was confirmed with PCR. The identity of the cell line was authenticated with STR profiling (FBI, CODIS). All the results can be viewed on the website (http://cellresource.cn). Transfections were performed using the polyethylenimine method. To establish human SEC14L2 knockdown stable cell line, negative control shRNA (SHC016) and SEC14L2 shRNA (TRCN0000019589, TRCN0000019590) plasmids (ordered from Sigma) were transfected into HEK293T/PC3 cells and then transfected cells were selected with puromycin (1 µg/ml). Following removal of the puromycin, the cells were allowed to recover and expand in regular growth medium and then were screened by protein immunoblotting. For serum starvation, cells were incubated in culture media in the absence of any additional supplements for 16 hr.

## Imaging of calcium levels in embryos and cytosolic calcium measurements by flow cytometry

To image calcium levels in embryos, pXT7-GCaMP6 plasmid was constructed based on pGP-CMV-GCaMP6 and linearized for GCaMP6 mRNA labelling. Then GCaMP6 mRNA was mixed with Rhodamine and injected into 1 cell stage embryos, which were embed in low-melting agar at the indicated stages for time lapse imaging. Then the resulting images were treated to generate pseudocolor ratio images as previously described (*Slusarski et al., 1997b*, *1997a*).

For calcium measurements in PC3 cells, cells were transfected with indicated plasmids for three days and loaded with calcium dye Quest Fluo 8-AM (AAT Bioquest, Sunnyvale, CA, 21083) or Rhod2-AM (AAT 21064) before flow cytometry analysis. During analysis, baseline fluorescence was measured for 50 s and stopped for Wnt5a addition, then measurement was resumed immediately for a total of 200s.

## Western Blot, co-immunoprecipitations, immunostaining and GST pull-downs

Western blots, co-IPs and immunostaining were performed as previously described (*Zhang et al., 2009*). For two-step of Co-IPs, the first step Co-IP components were eluted with the reduced glutathione for the second step immunoprecipitation experiments. Staining of PIP$_2$ in the PM was carried out according to a protocol previously published by *Hammond et al. (2009)*. Fluorescent images were acquired using a Nikon A1RMPsi lasers scanning confocal microscope. The following commercial antibodies were used in this study: anti-Flag (F1804, Sigma, RRID:AB_262044), anti-HA (sc-7392, Santa Cruz, Dalls, TX, RRID:AB_627809), anti-Myc (sc-40AC, Santa Cruz, RRID:AB_627268), anti-GFP (sc-9996, Santa Cruz, RRID:AB_627695) and anti-hDVL2 (#3216, CST, RRID:AB_2093338). Wnt3a (5036-WN), Wnt5a (645-WN) and Wnt11 (6179-WN) ligands were purchased from R&D (Minneapolis, MN). For GST pull-down assay, GST-Plcδ4a-HA and GST-Sec14l3 fusion proteins were expressed in *E.coli* and purified using glutathione-Sepharose beads (GE Healthcare, Marlborough, MA). After washing with PBS, GST-Plcδ4a-HA beads were incubated with PreScission Protease (GE Healthcare) to remove the GST tag and the resulting Plcδ4a-HA was harvested by PBS elution. The GST-Sec14l3 beads were washed with PBS and then incubated with the purified Plcδ4a-HA fusion protein for 2 hr at 4°C, and then washed with PBS again. The final eluent was analyzed by western blot using anti-GST and anti-HA antibodies. For GTPγS/GDP-Sec14l3 pull down assay, immobilized GST-Sec14l3 protein (on glutathione-Sepharose beads) was prepared and incubated with binding buffer (50 mM Tris-HCl [pH 7.4], 100 mM NaCl, 0.4% [vol:vol] Nonidet P-40, 10 mM MgCl$_2$, 5 mM EDTA, 30 μM GTPγS /GDP, 2 mM DTT, protease inhibitor mixture) for 90 mins at room temperature as described before (*Wu et al., 1993*; *Aznar et al., 2015*). Then lysates of HEK293T cells with hFz5-Myc or mDvl2-Flag plasmid transfection or purified Plcδ4a-HA (5 μg) protein were added and rotated at 4°C for another 2 hr. Beads were then washed using wash buffer (4.3 mM Na$_2$HPO4, 1.4 mM KH$_2$PO$_4$ [pH 7.4], 137 mM NaCl, 2.7 mM KCl, 0.1% [vol:vol] Tween 20, 10 mM MgCl$_2$, 5 mM EDTA, 30 μM GTPγS /GDP, 2 mM DTT) for 4 times every 3 min, finally boiled in 2xloading buffer for SDS-PAGE using corresponding antibodies.

## Liposome binding and membrane isolation assays

For the liposome binding assay, 1 μg HA-tagged Plcδ4a protein purified from *E. coli*, 20 μl 1 mM PolyPIPosomes (Echelon Biosciences, Salt Lake City, UT, Y-0000 and Y-P045), none or 5% PIP$_2$, 500 μl cell lysates transfected with control shRNA or *SEC14L2* shRNA and 500 μl binding buffer (50 mM Tris, pH 7.5, 150 mM NaCl, 0.05% NonidetP-40) were mixed and rotated for 4 hr at 4°C and then centrifuged at 13,000 rpm for 10 min. The liposome pellet was then washed with 1 ml of binding buffer (50 mM Tris, pH 7.5, 150 mM NaCl, 0.05% NonidetP-40) for three times. The bound and flow-through samples were eluted in 2xSDS loading buffer, separated by SDS-PAGE, transferred to nitrocellulose, and then Plcδ4a levels were measured by immunoblot with anti-HA antibody.

For the membrane isolation assay, a Minute$^{TM}$ Plasma Membrane Protein Isolation Kit (Invent Biotechnologies, Inc., Plymouth, MN, Catalog number: SM-005) was used according to the manufacturer's instructions. The isolated membrane samples were then dissolved in 2xSDS loading buffer and detected by separated by western blot using appropriate antibodies.

## GTP binding and hydrolysis assays

GTP binding and hydrolysis activity are measured using BODIPY-FL-GTPγS conventional assay and MESG-based single-turnover assay respectively; BODIPY-FL-GTPγS conventional assay is based on the release of the fluorescence quenching of BODIPY-FL-GTPγS (a non-hydrolyzable GTP analog) upon its binding to G proteins. BODIPY-FL-GTPγS (Invitrogen, G22183) binding to recombinant Sec14l3 was determined in 10 mM HEPEs (pH 8.0), 1 mM EDTA and 10 mM $MgCl_2$ (HEM buffer). The fluorescence ($\lambda_{ex}$ = 490 nm and $\lambda_{em}$ = 510 nm) was monitored for samples at different concentrations of BODIPY-FL-GTPγS, following the addition of 10 μM Sec14l3 protein, in a fluorescence microplate reader (Thermo Scientific VARIOSKAN FLASH). For the kinetic experiments, fluorescence of BODIPY-FL-GTPγS alone was measured in HEM at room temperature for 3 min and then binding was initiated with addition of excess Sec14l3. The change in fluorescence was measured over time and fitted with one phase exponential equation: $a*(1-e^{-kt})$ to obtain the $K_{diss}$ constant using Graph-Pad Prism 5.

The hydrolysis of GTP by Sec14l3 was measured by the MESG system monitoring the time course absorbance at 360 nm. The reaction was determined in 100 μl solution containing 50 mM MOPs (pH 7.0), 1 mM EDTA, 200 mM GTP, 1 U/ml purine nucleoside phosphorylase, 0.2 mM MESG and the recombinant Sec14l3 protein (reconstituted in Tris-HCl buffer). Single-turnover GTPase reactions were initiated by adding of $MgCl_2$ to a final concentration of 5 mM using the injector unit followed by immediate measurement (Thermo Scientific VARIOSKAN FLASH). The hydrolysis data fitted with one phase exponential equation: $a*(1-e^{-kt})$ to obtain the $K_{hydr}$ constant using GraphPad Prism 5. The time-course of Pi release from Sec14l3-GTP at four different concentrations was monitored, and finally averaged to determine the $K_{hydr}$ constant of Sec14l3.

For measurements of Plcδ4a GAP functions, indicated amounts of GAP proteins were mixed with Sec14l3 before $MgCl_2$ initiation, and control experiments with indicated GAP proteins were carried out to provide a background of absorbance in each independent measurement to be subtracted from the sample signals.

Steady-state GTPase activity was performed using a QuantiChrom ATPase/GTPase Assay kit (GENTAUR, San Jose, CA, DATG-200) according to the manufacturer's instructions.

## Statistical analysis

Quantitative data are presented as mean ± SEM, and comparisons were performed between groups using a two-tailed Student's t-test. For all analyses, *p<0.05; **p<0.01 were considered statistically significant; ns indicated statistical non-significance with p>0.05. Each experiment was carried out at least three times independently.

## Acknowledgements

We thank Jiawei Wu and Jue Wang for help with MESG-based single-turnover assay, Wei Wu for providing hFz5-Myc, Bailong Xiao for STIM1-mCherry, Douglas Kim for pGP-CMV-GCaMP6 (Addgene plasmid #40753) and Randall Moon for XE128 Rfz2 TG2myc CS2+ (Addgene plasmid #16796). We are grateful to the members of the Meng Laboratory and Drs. Qiang Wang and Wei Wu for helpful discussion and technical assistance. This work was financially supported by grants from the National Natural Science Foundation of China (#31522035 and #31371460), Ministry of Science and Technology of the People's Republic of China (#2012CB945100, #2011CB943800).

## Additional information

### Funding

| Funder | Grant reference number | Author |
| --- | --- | --- |
| National Natural Science Foundation of China | 31522035 | Shunji Jia |
| National Natural Science Foundation of China | 31371460 | Shunji Jia |
| Ministry of Science and Tech- | 2012CB945100 | Shunji Jia |

| | | |
|---|---|---|
| nology of the People's Republic of China | | |
| Ministry of Science and Technology of the People's Republic of China | 2011CB943800 | Anming Meng |

The funders had no role in study design, data collection and interpretation, or the decision to submit the work for publication.

## Author contributions

BG, Data curation, Software, Formal analysis, Validation, Investigation, Methodology, Writing—original draft; WS, Software, Visualization, Methodology; WX, Resources, Visualization, Methodology; YM, Visualization, Methodology; AM, Conceptualization, Supervision, Validation, Project administration, Writing—review and editing; SJ, Supervision, Funding acquisition, Validation, Methodology, Writing—original draft, Project administration, Writing—review and editing

## Author ORCIDs

Shunji Jia, http://orcid.org/0000-0001-8678-1714

## Ethics

Animal experimentation: Fishes were handled according to the institutional animal care and use committee (IACUC) protocols (AP#13-MAM1), which were approved and permitted by the Tsinghua University Animal Care and Use Committee.

## Additional files

### Supplementary files

• Supplementary file 1. Primers used in the present study.

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
