## [Decision Letter]

Thank you for submitting your article "The Sec14-like Phosphatidylinositol Transfer Proteins Act as GTPase Proteins to Mediate Wnt/Ca^2+^ Signaling" for consideration by *eLife*. Your article has been favorably evaluated by Didier Stainier (Senior Editor) and three reviewers, one of whom, Hong Zhang (Reviewer #1), is a member of our Board of Reviewing Editors. The following individual involved in review of your submission has agreed to reveal their identity: Weijun Pan (Reviewer #3).

The reviewers have discussed the reviews with one another and the Reviewing Editor has drafted this decision to help you prepare a revised submission.

Summary:

This manuscript provides a comprehensive analysis of the zebrafish and human phosphatidylinositol transfer protein Sec14l3/SEC14L2 in the activation of the Wnt/Ca^2+^ signaling pathway. The authors show that Wnt5a stimulation induces the membrane recruitment of Sec14l3. The GDP-bound Sec14l3 has the potential to form a complex with Frizzled and Dishevelled, while Sec14l3-GTP preferentially binds and activates Plcδ4a to trigger calcium release through PIP2 signaling. The results suggest that Sec14l3 functions as a GTPase protein and is specifically required for Wnt/Ca^2+^ signaling, but not for Wnt/PCP signaling. Functional analyses in zebrafish indicate that maternal Sec14l3 is required for convergence and extension movements. Overall, these findings provide new mechanistic insights into the regulation and functional implication of the Wnt/Ca^2+^ pathway in cell movements during early development. They will be of great interest in the Wnt field. This manuscript is suitable for publication in *eLife* after appropriate revisions.

Essential revisions:

1) Although Wnt5a stimulation or Frizzled/Dishevelled overexpression promotes Sec14l3 membrane recruitment, it is not clear whether this could switch the conformational change of Sec14l3, from the GDP-bound state to the GTP-bound state. Dishevelled binds preferentially GDP-bound Sec14l3, but there is no evidence showing that this interaction is necessary to convert Sec14l3 into an active state. Whether the formation of the Fz/Dvl/Sec14l3 complex is required to activate Sec14l3 needs to be addressed or discussed properly.

2) Sec14l3 utilizes distinct domains for physical interaction with Dvl and Plcδ4a, but which Sec14l3 domain interacts with the Fz C-terminal region is not clear. Except for the two conserved motifs (KTxxxW and S/TxV), the C-terminal region is very divergent between hFz5 and RFz2, and the RFz2 has a very short C-terminal region (26aa). It will be interesting to know how the Fz C-terminal region interacts with Sec14l3, or at least this aspect should be discussed.

3) Both knockdown and knockout of Sec14l3 affects convergence and extension movements, but Sec14l3 morphants and mutants also displays other early developmental defects, in particular, a very strong epiboly delay (see Figure 2). So, the possibility of an impaired cell adhesion between EVL and deep cells during epiboly cannot be excluded and needs to be discussed.

4) The interaction between Wnt/Ca^2+^ and Wnt/PCP pathways in convergence and extension needs to be discussed.

---

## [Author Response]

Essential revisions:

1) Although Wnt5a stimulation or Frizzled/Dishevelled overexpression promotes Sec14l3 membrane recruitment, it is not clear whether this could switch the conformational change of Sec14l3, from the GDP-bound state to the GTP-bound state. Dishevelled binds preferentially GDP-bound Sec14l3, but there is no evidence showing that this interaction is necessary to convert Sec14l3 into an active state. Whether the formation of the Fz/Dvl/Sec14l3 complex is required to activate Sec14l3 needs to be addressed or discussed properly.

We agree with the reviewers, and so far we have not provided direct evidences to support the conformational change of Sec14l3 upon Wnt5a stimulation or Fz/Dvl overexpression, but only some discussion in our initial submission version. One major obstacle is the lack of specific antibodies, which can recognize and distinguish Sec14l3-GDP from Sec14l3-GTP state. Considering that Dvl and Plcδ4a prefer to bind with GDP- and GTP- form of Sec14l3 respectively (Figure 6), and GTPase activity of Sec14l3 is essential for activating Plcδ4a at the plasma membrane (Figure 4, Figure 5 and Figure 6). We propose membrane-localized Sec14l3 somehow represents its active state, in GTP-bound form, while cytosol-localized Sec14l3 might be inactive, in GDP-bound state. As Wnt5a stimulation and Fz/Dvl overexpression can promote Sec14l3 membrane recruitment, it is likely that Fz/Dvl/Sec14l3 complex is required to activate Sec14l3. And actually, we have proved the existence of Fz/Dvl/Sec14l3 complex using sequential co-immunoprecipitation approach. In our experiments, HEK293T cell lysates transfected with Dvl2-Flag was incubated with GST-Sec14l3/GST and hFz5-CT-Myc proteins, and GST-Sec14l3/GST was immunoprecipitated (1st step co-IP). The co-IP components were then eluted with the reduced glutathione and immunoprecipitations were carried out for the second round using anti-Flag antibody (2nd step co-IP). Results showed that hFz5-CT-Myc was only found in the final elute of GST-Sec14l3 group, but not in the GST control group, indicating that Fz/Dvl/Sec14l3 are present in the same complex (New data, Figure 2). Certainly, it will be more convincing if we could compare conformations of Sec14l3 in the complex or alone, when their structures are available. The new data have been included in our revised manuscript.

2) Sec14l3 utilizes distinct domains for physical interaction with Dvl and Plcδ4a, but which Sec14l3 domain interacts with the Fz C-terminal region is not clear. Except for the two conserved motifs (KTxxxW and S/TxV), the C-terminal region is very divergent between hFz5 and RFz2, and the RFz2 has a very short C-terminal region (26aa). It will be interesting to know how the Fz C-terminal region interacts with Sec14l3, or at least this aspect should be discussed.

We appreciate reviewers for this comment. To further explicate the interaction between Fz C-terminal region and Sec14l3, we mapped Sec14l3 domains after co-transfection of distinct Sec14l3 truncated forms with Rfz2-CT. It revealed that the N-terminal CARL-TRIO and C-terminal GOLD2 domains of Sec14l3 are required for this interaction (New data, Figure 2—figure supplement 1). This new data have been included in our revised manuscript.

Based on the fact that these two Fz C-terminals both interact with Sec14l3 and only these two conserved motifs (KTxxxW and S/TxV) along them, we’d like to speculate that these two conserved motifs may be also critical for the complex formation. To investigate how the Fz C-terminal region interacts with Sec14l3 in detail, more direct evidences could be acquired from isothermal titration calorimetry (ITC) experiments, using synthetic peptides from Fz C-terminal region and purified Sec14l3 proteins. Furthermore, membrane phospholipids may also cooperate with Fz-Sec14l3 interaction to stabilize the complex at the plasma membrane, because Sec14l3 also can act as a lipid binding protein (data not shown). If proven true, the interaction between Fz and Sec14l3 will be quite similar with that of Fz and Dvl, which is also dependent on these two conserved motifs, as well as membrane acidic phospholipids (Simons et al., 2009, Wong et al., 2003, Umbhauer et al., 2000, Hering and Sheng, 2002).

3) Both knockdown and knockout of Sec14l3 affects convergence and extension movements, but Sec14l3 morphants and mutants also displays other early developmental defects, in particular, a very strong epiboly delay (see Figure 2). So, the possibility of an impaired cell adhesion between EVL and deep cells during epiboly cannot be excluded and needs to be discussed.

We appreciate reviewers for this comment. In order to examine the exact marginal positions of EVL and deep cells in *sec14l3* knockdown or knockout embryos, we performed immunostaining using phalloidin and DAPI. Our analyses indicated that, compared with control embryos, there is no significant difference in the distance between the marginal borders of the EVL and the deep cell layer in *sec14l3*morphant or M*sec14l3* mutant embryos (New data, Figure 1—figure supplement 2). Therefore, it seems that Sec14l3 has no obvious effect on cell adhesion between EVL and deep cells. We included this new data in our revised manuscript.

4) The interaction between Wnt/Ca^2+^ and Wnt/PCP pathways in convergence and extension needs to be discussed.

We appreciate reviewers for this comment. Wnt/Ca^2+^ and Wnt/PCP signaling are classified into the non-canonical Wnt signaling, which is independent of β-catenin transduction. Although it is not yet clear to what extent the Wnt/Ca^2+^ and PCP pathways overlap, several Wnts or Fz receptors/co-receptors can trigger both of them according to distinct contexts. Therefore, these two pathways are both devoted to modulate cytoskeleton organization to coordinate or polarize cell convergent and extension movements (Veeman et al., 2003, Yang and Mlodzik, 2015, De, 2011, Kohn and Moon, 2005). However, it seems that Wnt/Ca^2+^ functions through more possible ways, including regulating calcium-dependent cell adhesion, dynamics of calcium storage and release, or modulating both PCP and Wnt/β-catenin pathways (Kuhl et al., 2001, Slusarski and Pelegri, 2007, Tada and Heisenberg, 2012, Wallingford et al., 2002). In our case, knocking down *sec14l3* attenuates calcium release, but not phosphorylation level of *Jnk*, somehow indicating that Sec14l3 affects cell convergent extension movement mainly through regulating calcium waves. We introduced this part properly in our new version.